# Epsilon tubulin is an essential determinant of microtubule-based structures in male germ cells

G Gemma Stathatos [1,2], D Jo Merriner[1], Anne E O'Connor [1], Jennifer Zenker [2], Jessica EM Dunleavy [1]✉ & Moira K O'Bryan [1]✉

## Abstract

**Alpha, beta, and gamma tubulins are essential building blocks for all eukaryotic cells. The functions of the non-canonical tubulins, delta, epsilon, and zeta, however, remain poorly understood and their requirement in mammalian development untested. Herein we have used a spermatogenesis model to define epsilon tubulin (TUBE1) function in mice. We show that TUBE1 is essential for the function of multiple complex microtubule arrays, including the meiotic spindle, axoneme and manchette and in its absence, there is a dramatic loss of germ cells and male sterility. Moreover, we provide evidence for the interplay between TUBE1 and katanin-mediated microtubule severing, and for the sub-specialization of individual katanin paralogs in the regulation of specific microtubule arrays.**

**Keywords** Epsilon Tubulin; Spermatogenesis; Meiosis; Manchette; Axoneme
**Subject Categories** Cell Adhesion, Polarity & Cytoskeleton; Development

## Introduction

The tubulin superfamily are the building blocks of the microtubule cytoskeleton. The canonical alpha (α) and beta (β) tubulin subunits constitute the fundamental structure of microtubule filaments, and alongside the microtubule nucleator, gamma (γ) tubulin (Janke and Magiera, 2020; Kollman et al, 2011), are conserved across all eukaryotes. Three additional non-canonical and poorly defined tubulins exist, termed the ZED-tubulins—zeta (ζ), epsilon (ε, TUBE1), and delta (δ, TUBD1) tubulin (Findeisen et al, 2014; Stathatos et al, 2021). ZED-tubulin genes are found in species wherein centrioles comprised of triplet microtubules are present (Turk et al, 2015). In species where function has been tested, largely unicellular flagellates and cell lines, centriole triplet microtubules promote centriole stability, inheritance and are almost always required for the development of motile cilia (Carvalho-Santos et al, 2011; Wang et al, 2017). Interestingly, TUBE1 is always present

alongside TUBD1 and/or ζ-tubulin, suggesting a core TUBE1 function that can be supplemented by either ζ-tubulin or TUBD1 (Turk et al, 2015). Humans and mice, for example, possess TUBD1 and TUBE1, but not ζ-tubulin, whereas unicellular species, such as *Bigelowiella natans*, possess TUBE1 and ζ-tubulin, but not TUBD1 (Turk et al, 2015). TUBE1 and TUBD1 have also shown to be capable of interacting in human embryonic kidney cells (Wang et al, 2017). In summary, data thus far hints at evolutionary conserved roles for the ZED-tubulins at the centriole and centriole-derived organelles—the centrosome, the basal body, and cilia/flagella (Stathatos et al, 2021). However, this hypothesis has not been tested in vivo in mammals.

The structure of TUBE1 is yet to be determined, however, like the canonical tubulins at a sequence level, it possesses a nucleotide-binding domain for 'head-to-tail' tubulin binding, and a GxxNxD amino acid sequence associated with tubulin-tubulin lateral binding (Inclán and Nogales, 2001; Löwe et al, 2001). It is also predicted to interact with α-/β-tubulins via select longitudinal and lateral axes (Inclán and Nogales, 2001; Stathatos et al, 2021). Unlike α- and β-tubulin, however, TUBE1 lacks the acidic C-terminal tail that is a primary target of post-translational modifications (Inclán and Nogales, 2001; McKean et al, 2001) and canonical microtubule severing enzymes (Zehr et al, 2020).

The handful of studies conducted on TUBE1 thus far have consistently highlighted roles at the centrosome and its derivative, the basal body, and thus ciliogenesis/flagellogensis (reviewed in Stathatos et al (2021)). For example, TUBE1 has been localized to the centrosome and sub-distal appendages in human bone osteosarcoma epithelial (U2OS) cells (Chang and Stearns, 2000), and the pericentrosomal area of the basal body in elongating mouse spermatids; the structure from which the axoneme, the microtubule core of the sperm tail, develops (Dunleavy et al, 2017). Loss of TUBE1 function in flagellates and ciliates, including *Chlamydomonas reinhardtii* (Dutcher et al, 2002), *Tetrahymena thermophila* (Ross et al, 2013), and *Paramecium tetraurelia* (Dupuis-Williams et al, 2002), resulted in the absence of the outer (B- and C-) tubules from the centriole microtubule triplets of the basal body (Dupuis-Williams et al, 2002; Ehler et al, 1995; Ross et al, 2013). Specifically, the loss of TUBE1 in *C. reinhardtii* resulted in an absence of flagella in mutant organisms (Dutcher et al, 2002). Similarly, in mutant *P. tetraurelia*, cilia were rare, however, when present axoneme ultrastructure was overtly normal (Dupuis-Williams et al, 2002).

[1]School of BioSciences and Bio21 Institute of Molecular Science and Biotechnology, Faculty of Science, The University of Melbourne, Parkville, VIC 3010, Australia. [2]Australian Regenerative Medicine Institute, Monash University, Clayton, VIC 3800, Australia. ✉E-mail: jessica.dunleavy@unimelb.edu.au; moira.obryan@unimelb.edu.au

In regards to the functioning of the centriole proper, *TUBE1* loss in p53-null human retinal pigment epithelial cells (RPE1) led to centriole disintegration, which triggered de novo centriole formation during each cell cycle (Atorino et al, 2020; Wang et al, 2017).

While data thus far paints a picture of an evolutionarily conserved role for TUBE1 in centrioles and centriole-based organelles and processes, this remains to be studied in mammals in vivo in either somatic cells, or in germ cells. Within mammals, the enrichment of *Tube1* expression in male germ cells (Dunleavy et al, 2017) suggests TUBE1 may be especially important for spermatogenesis, a process which is heavily reliant on a number of complex microtubule structures (Dunleavy et al, 2019a). Through immunolabeling of testis sections and haploid male germ cells, TUBE1 has been localized to the manchette and, as detailed above, the basal body (Dunleavy et al, 2017). The manchette is a transient microtubule-based array, which forms a skirt-like structure around the spermatid nucleus wherein microtubules are anchored to a perinuclear ring 'waist-band' (Russell et al, 1991). The manchette mechanically shapes the distal half of the sperm head by constricting while progressively ratcheting down the nucleus via the dynamic 'unzippering' and 're-zippering' of microtubule-nuclear and perinuclear ring-nuclear (together manchette-nuclear) linker complexes (Dunleavy et al, 2019a; Russell et al, 1991). In addition, the manchette serves as a transport platform for proteins and organelles required for sperm tail assembly (Dunleavy et al, 2017; Pleuger et al, 2020).

In regard to manchette movement and sperm head shaping, growing evidence supports the involvement of linkers of nucleoskeleton and cytoskeleton (LINC) complex members SUN3 and SUN4 as key components of the nuclear complex that anchors to the linkers spanning between the inner microtubules of the manchette and the nucleus (Calvi et al, 2015; Gao et al, 2020; Pasch et al, 2015). The composition of each of the microtubule-nuclear and perinuclear ring-nuclear linkers and the mechanism by which the ratcheting of the manchette down the spermatid nucleus is achieved is unknown. The failure of manchette migration in the absence of the katanin genes *Katnal2* or *Katnal1*, or their regulatory B-subunit *Katnb1* (Dunleavy et al, 2023; Dunleavy et al, 2021; Dunleavy et al, 2017) does, however, imply a role for microtubule severing in this process. Notably, while the composition of the links is unknown, TUBD1 has been localized to the perinuclear ring (Kato et al, 2004) and we have identified TUBE1 as an interacting partner of the microtubule severing protein—KATNAL2 (katanin-like 2) at the manchette and pericentriolar region of elongating spermatids (Dunleavy et al, 2017). KATNAL2 loss leads to a failure of manchette migration and an absence of axoneme/tail formation (Dunleavy et al, 2017), however, its mechanism of action is currently unknown as it does not appear to sever α-β-tubulin-based microtubules in the same way other katanin A subunits do (Cheung et al, 2016; Dunleavy et al, 2017; Ververis et al, 2016). We thus hypothesized that TUBE1 is an attractive candidate component of the manchette-nuclear linkers severed by KATNAL2.

Within this study, we sought to test the requirement for TUBE1 in a mammalian developmental process in vivo, by characterizing the function of TUBE1 in male germ cell development and fertility. Moreover, we sought to directly test our hypothesis that TUBE1 is a key target of KATNAL2 microtubule severing. Using a conditional germ cell-specific knockout mouse model (*Tube1^GCKO/GCKO*), we show that TUBE1 is essential for male fertility, including for

meiotic bipolar spindle assembly, chromosome segregation and cytokinesis. In haploid germ cell development, TUBE1 is required for functional axoneme assembly and stability and nuclear remodeling via the manchette. In addition, we provide evidence that TUBE1 functions in a complex with the katanin microtubule severing proteins to mediate manchette movement during the process of nuclear remodeling. We also provide evidence for the neo-functionalization of katanin A-subunits in the severing of key microtubule arrays during spermatogenesis.

# Results

## TUBE1 is required for male germ cell development and fertility

To explore the role of TUBE1 during male fertility, a germ cell-specific knockout mouse model (*Tube1^GCKO/GCKO*) was produced using *Stra8-Cre* for pre-meiotic deletion of exon 3 from *Tube1* isoforms in the male germline (Fig. EV1A). There are two transcribed *Tube1* isoforms in the mouse, however, only one isoform is expected to produce functional protein (*Tube1-201*), as the second undergoes nonsense-mediated decay (*Tube1-203*) (Fig. EV1B). *Tube1* mRNA was reduced by 90.1% in purified *Tube1^GCKO/GCKO* spermatocytes compared to *Tube1^Flox/Flox* controls, thus confirming efficient deletion (EV1C). Noting that such preparations are on average 81% pure, the remnant mRNA detected in these purifications is likely attributed to contaminating somatic cells or spermatogonia (Dunleavy et al, 2019b). *Tube1^GCKO/GCKO* male mice were outwardly normal, had normal body weight (Fig. EV1D), and displayed normal mating frequency, but were sterile (Fig. 1A). In accordance, *Tube1^GCKO/GCKO* adult testis weights were reduced by 64.7% (Figs. 1B and EV2A), daily sperm production (DSP) by 80.5% (Fig. 1C) and total epididymal sperm content by 91.6% (Fig. 1D) compared to *Tube1^Flox/Flox* controls. The more severe reductions in epididymal sperm content, in comparison to DSP, is indicative of spermiation (sperm release) failure. Histological examination confirmed this, showing spermatozoa were abnormally retained within the *Tube1^GCKO/GCKO* seminiferous epithelium at Stage IX (Fig. 1E).

At a histological level, most seminiferous tubule cross-sections of *Tube1^GCKO/GCKO* mice displayed hypospermatogenesis (Figs. 1F and EV2B; 60.18%, compared to 3.42% in controls) in which all male germ cell types were present, but at reduced levels compared to those from *Tube1^Flox/Flox* controls. In addition, 28.07% of *Tube1^GCKO/GCKO* tubules showed a Sertoli cell-only (SCO) phenotype in which no germ cells could be seen (Fig. 1F, black arrowhead, and Fig. EV2B), compared to 0.28% in control mice (*Tube1^Flox/Flox*). Furthermore, Sertoli cell symplasts were occasionally observed in *Tube1^GCKO/GCKO* tubules (Fig. 1F, red arrowhead). In accordance, staining for cleaved Caspases 3 and 7 revealed a significant increase in the number of apoptotic germ cells per seminiferous tubule in *Tube1^GCKO/GCKO* mice compared to control mice (Fig. EV2C), with positive staining most frequently seen at the level of pachytene and metaphase spermatocytes and spermatogonia (Fig. EV2D). In addition, examination of epididymal histology revealed the presence of round cells in the epididymal lumen of *Tube1^GCKO/GCKO* mice, indicative of precocious germ cell sloughing, which was not observed in controls (Fig. EV2E).

Although sperm present in the *Tube1^GCKO/GCKO* epididymis possessed tails, they were not motile, revealing that TUBE1 is required to form a motility-competent sperm flagellum. In the absence of TUBE1, 14.7%

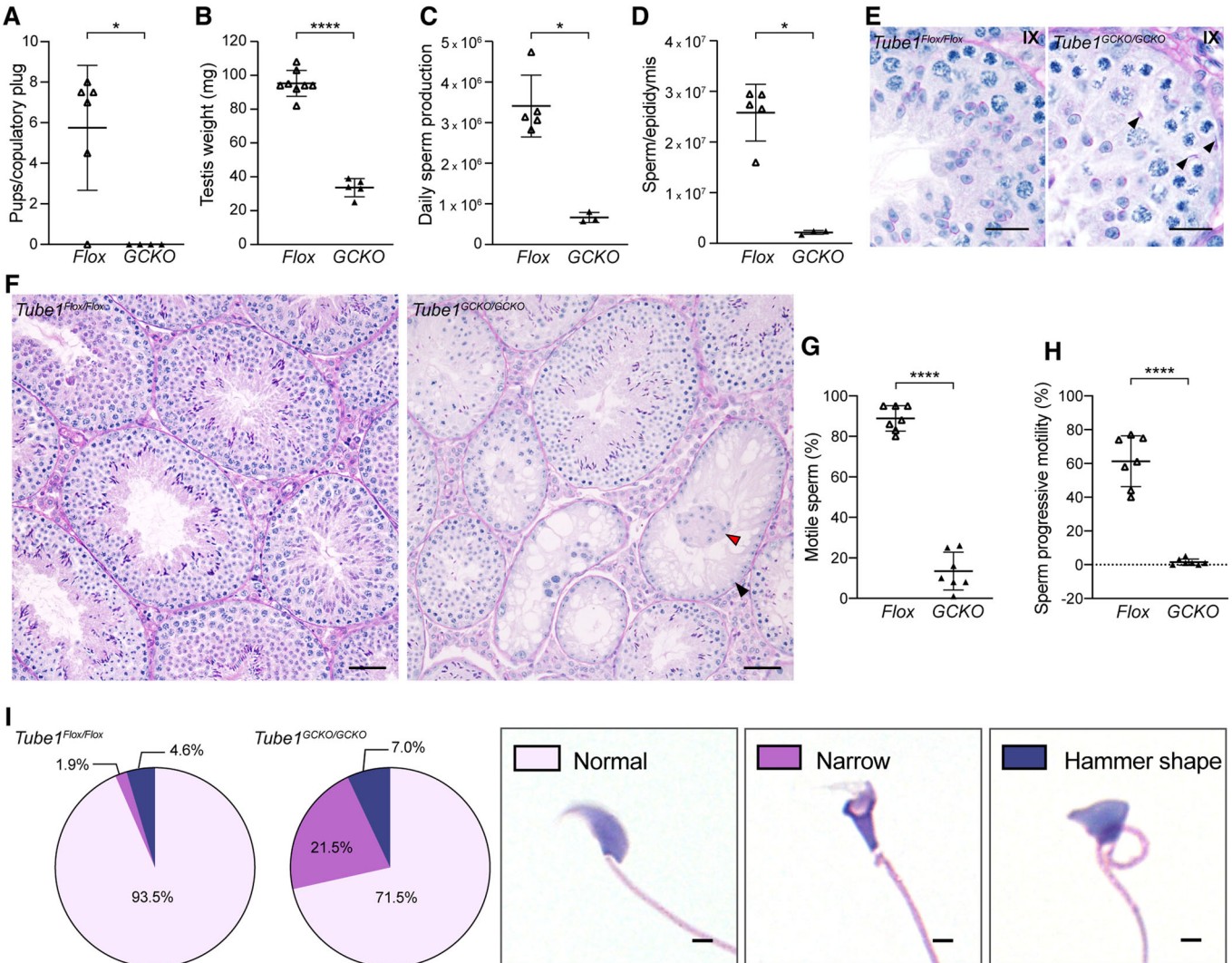

**Figure 1. The pre-meiotic loss of TUBE1 results in sterility in male mice.**

(A) The average number of pups born from *Tube1^Flox/Flox^* (*Flox*) and *Tube1^GCKO/GCKO^* (*GCKO*) mice ($n = 6$ *Flox* and $n = 4$ *GCKO* males, 2 technical replicates; Mann–Whitney U test, $P = 0.0333$, *). (B) Testis weight in *Tube1^Flox/Flox^* and *Tube1^GCKO/GCKO^* mice ($n = 8$ *Flox* and $n = 5$ *GCKO* animals, 1 technical replicate; unpaired t-test, $P < 0.0001$, ****). (C, D) Daily sperm production (C) and total epididymal sperm count (D) in *Tube1^Flox/Flox^* and *Tube1^GCKO/GCKO^* mice ($n = 5$ *Flox* and $n = 3$ *GCKO* animals, 2 technical replicates; Mann–Whitney U test, $P = 0.0357$, *). (E) Cross sections of Periodic acid Schiff's (PAS)-stained stage IX seminiferous tubules, with *Tube1^GCKO/GCKO^* displaying retained sperm heads (black arrowheads) ($n = 3$ animals/genotype). Scale bars: 20 µm. (F) PAS-stained testis sections. Black and red arrowheads indicate Sertoli cell only tubules and symplasts, respectively ($n = 3$ animals/genotype). Scale bars: 50 µm. (G) Sperm motility in *Tube1^Flox/Flox^* and *Tube1^GCKO/GCKO^* mice ($n = 7$ animals/genotype, 1 technical replicate; unpaired t-test, $P < 0.0001$, ****). (H) Percentage of sperm displaying progressive motility in *Tube1^Flox/Flox^* and *Tube1^GCKO/GCKO^* mice ($n = 7$ animals/genotype, 1 technical replicate; unpaired t-test, $P < 0.0001$ (Welch's correction), ****). (I) Distribution of sperm head shapes present in epididymal sperm alongside representative hematoxylin and eosin stained epididymal sperm heads ($n = 3$ animals/genotype, 1 technical replicate). Scale bars: 2 µm. Data information: For each panel, results are from one experiment. In (A, D, G, H), data are presented as mean ± SD. Source data are available online for this figure.

of sperm displayed the capacity for motility, compared to 89.7% of sperm from *Tube1^Flox/Flox^* mice (Fig. 1G). Of these, only 2.3% manifested progressive (forward) motility, compared to 65.3% of sperm from *Tube1^Flox/Flox^* mice (Fig. 1H). To further test the underlying functionality of the axoneme, cell permeable ATP was added to sperm and motility monitored. The addition of ATP did not significantly boost the number of motile sperm or the mean sperm velocity of *Tube1^GCKO/GCKO^* mice (Fig. EV2F), indicating that deficits were integral to the axoneme rather than a consequence of insufficient ATP production. These data reveal that TUBE1 is required for the formation a functional axoneme in haploid mammalian male germ cells.

In addition to motility defects, 28.5% of sperm from *Tube1^GCKO/GCKO^* mice had abnormally shaped heads compared to 6.5% of sperm from *Tube1^Flox/Flox^* mice (Fig. 1I). The most common defect were 'narrow' heads, including those that exhibited a 'knobby' head shape, where the lower portion of the nucleus was considerably narrower than the upper portion (21.5% of sperm from *Tube1^GCKO/GCKO^* mice versus 1.9% of sperm from *Tube1^Flox/Flox^* mice). Of note, and as detailed below in the manchette function section, assessment of abnormal head shape morphology in spermatids prior to their release from the testis via spermiation revealed in *Tube1^GCKO/GCKO^* mice, elongated spermatids exhibited a higher percentage (50.6%) of abnormal head shapes than in

epididymal sperm. This suggests morphologically abnormal sperm are more likely to fail to undergo spermiation, potentially due to a quality control mechanism. Interestingly, examining the mRNA expression of TUBE1's binding partner, TUBD1, in round spermatids revealed a 56% reduction of *Tubd1* in *Tube1^{GCKO/GCKO}* populations suggesting that *Tube1* and *Tubd1* may be co-regulated (Fig. EV2G).

Collectively, these data reveal that TUBE1 is essential for male mouse fertility. It is required for the maintenance of germ cell number and the establishment of optimal sperm head morphology and sperm motility. The loss of TUBE1 leads to a cumulative 99.7% reduction in the number of functional sperm produced in the mouse. The equivalent clinical presentation in humans is oligoasthenoteratozoospermia (OAT)—combined deficits in each of sperm output, motility, and morphology.

## TUBE1 is a key determinant of spindle organisation and cytokinesis in male meiosis

Consistent with the elevated apoptosis of spermatocytes and reduced spermatogenic output, meiosis was overtly abnormal in *Tube1^{GCKO/GCKO}* mice from metaphase onwards. Immunolabelling testis sections with α-tubulin as a maker of microtubules revealed that the loss of TUBE1 resulted in clear meiotic spindle defects (Fig. 2A), while counting the number of leptotene and pachytene cells present in Stage IX tubules did not reveal a difference between *Tube1^{Flox/Flox}* and *Tube1^{GCKO/GCKO}* mice in prophase I (Fig. EV3A). In *Tube1^{Flox/Flox}* mice, 86.7% of metaphase spermatocytes exhibited normal spindle morphology (Fig. 2B) and tightly packed chromosomes aligned at the metaphase plate (Fig. 2A i, iii, iv). In contrast, only 35.1% of *Tube1^{GCKO/GCKO}* metaphase spermatocytes exhibited normal spindle morphology. Instead, in 27.4% of *Tube1^{GCKO/GCKO}* metaphase spermatocytes, chromosomes aligned along the metaphase plate but were widely dispersed (Fig. 2A v, blue arrowhead, Fig. 2B), resulting in wider metaphase plates (Fig. 2A vi). Chromosome misalignment, defined as the deviation of chromosomes away from the equatorial plane of the metaphase plate, was observed in 37.5% of metaphase cells (Fig. 2A vii–viii, red arrowheads, Fig. 2B) in testis sections of *Tube1^{GCKO/GCKO}* mice, compared to 6.4% of cells from littermate controls. Furthermore, *Tube1^{GCKO/GCKO}* mice displayed uneven chromosome segregation in some anaphase spermatocytes (Fig. 2A ii, black arrowhead), resulting in round spermatids with abnormally small and large nuclei in *Tube1^{GCKO/GCKO}* testis (Fig. EV3B). In addition to metaphase and anaphase defects, the loss of TUBE1 resulted in elevated numbers of binucleated spermatids compared to *Tube1^{Flox/Flox}* controls (Fig. 2C, black arrowhead), indicative of failed cytokinesis at the end of meiosis. Binucleated spermatids were observed in *Tube1^{GCKO/GCKO}* mice at a frequency of 1.95% versus 0.00% in *Tube1^{Flox/Flox}* mice ($n \geq 7$ Stage VIII seminiferous tubules/animal, $n = 3$ animals/genotype, $p = 0.004$).

To further assess the role of TUBE1 in establishing normal bipolar meiotic spindle morphology, spermatocytes were purified and stained with β-tubulin and centrin as markers of microtubules and centrioles, respectively (Fig. 2D–H). While leptotene spermatocyte microtubule architecture was comparable between groups (Fig. EV3C), in *Tube1^{GCKO/GCKO}* spermatocytes, 38.28% of metaphase spindles were overtly abnormal compared to 1.66% from *Tube1^{Flox/Flox}* animals (Fig. 2I), whereby abnormalities were defined as unipolar (notably, with >2 centrin foci), multipolar, and unfocused spindles (Fig. 2F–J), where centrin-labeled foci were dispersed throughout the cytoplasm and not clearly associated with sites of microtubule enrichment.

Centrin-less spindle poles were also observed in multipolar cells (Fig. 2G, yellow arrowhead). Consistent with this, co-labeling metaphase spermatocytes with centrin and γ-tubulin revealed a higher proportion of cells from *Tube1^{GCKO/GCKO}* mice (46.7%) had more or less than 2 centrosomes, compared to those from *Tube1^{Flox/Flox}* mice (5.6%) (Fig. EV3D). Together, these data reveal a role for TUBE1 in regulating meiotic spindle polarity and for maintaining normal centrosome number in cells, either by suppressing supernumerary centriole formation, or by facilitating cell division and the appropriate separation of cell components, including centrioles into daughter cells. Resultingly, this ensures even chromosome separation and the formation of male meiotic spindles with correct bipolar morphology.

Concordant with the increase in cleaved-Caspase-positive cells in *Tube1^{GCKO/GCKO}* (Fig. EV2D), analysis of Periodic acid Schiff's-stained (PAS) testis sections revealed pyknotic spermatocytes stalled in metaphase and anaphase persisted beyond the normal completion of meiosis in stage XII into stage I seminiferous tubules in *Tube1^{GCKO/GCKO}* (Fig. 2K, black arrowheads). This was infrequently seen in *Tube1^{Flox/Flox}* testes.

## TUBE1 is not required for sperm axoneme initiation in mouse male germ cells but is required for axoneme stability and function

As established in the motility assays, the axonemes of *Tube1^{GCKO/GCKO}* mouse sperm were non-functional. To explore the origins of this defect, we characterized the initiation of axoneme formation in round spermatids by transmission electron microscopy (TEM). Consistent with the presence of sperm tails in *Tube1^{GCKO/GCKO}* mice (Fig. 1I), basal body docking to the nuclear and plasma membranes was observed in spermatids from both *Tube1^{Flox/Flox}* and *Tube1^{GCKO/GCKO}* mice (Fig. 3A). In some cases, however, the nuclear membrane docking site was ectopically shifted away from the caudal pole and towards the acrosome in *Tube1^{GCKO/GCKO}* spermatids (Fig. 3A, red arrowhead). Rarely, and suggestive of supernumerary centriole formation, sperm from *Tube1^{GCKO/GCKO}* mice presented with two tails (Fig. 3B, black arrowheads). This was not observed in sperm from *Tube1^{Flox/Flox}* mice.

Next, the ultrastructure of developing spermatid axonemes was examined—noting low numbers of epididymal sperm precluded an assessment of spermatozoa ultrastructure. Cross-sections of the tail midpiece, principal piece, and end piece in developing spermatids of *Tube1^{Flox/Flox}* mice displayed a canonical nine-fold array of the microtubule doublets and central pair (CP) within the axoneme (Fig. 3C). In *Tube1^{GCKO/GCKO}* mice, axonemes were observed, however, their structure ranged from superficially normal to severely abnormal (Fig. 3C,D). Abnormalities included off-center CPs (Fig. 3D i, red arrowhead), missing axonemal doublets (Fig. 3D ii, yellow asterisk), the loss of radial symmetry (Fig. 3D iii, blue arrowhead), and in the most severe instances complete disorganization of doublets and outer dense fibers (Fig. 3D iv). As sperm within the testis are immotile, these abnormalities are primary developmental defects rather than being secondary to the induction of sperm motility.

## TUBE1 is required for manchette function during nuclear remodeling

The identification of 'knobby' shaped sperm heads (Fig. 1I) suggested disrupted manchette dynamics in *Tube1^{GCKO/GCKO}* mice. To explore a

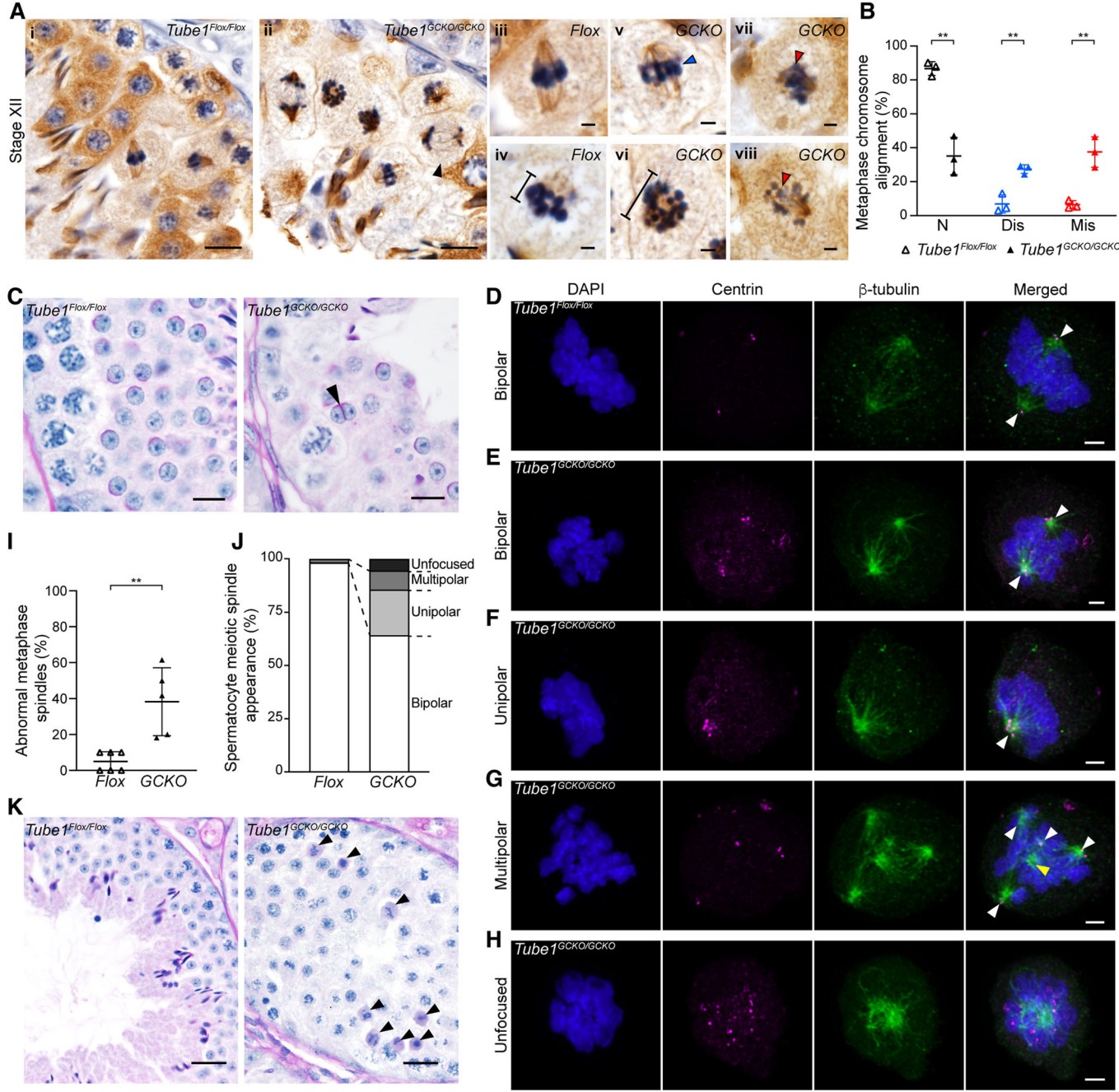

**Figure 2. TUBE1 is involved in multiple aspects of mammalian male meiosis.**

(A) Immunohistochemical staining of α-tubulin in Stage XII seminiferous tubule testis sections from *Tube1*<sup>Flox/Flox</sup> (*Flox*) and *Tube1*<sup>GCKO/GCKO</sup> (*GCKO*) mice (i–ii). Uneven chromosome segregation indicated by black arrowhead. Comparison of metaphase spermatocytes between genotypes (iii–viii) illustrate dispersed chromosome alignment (v, blue arrowhead), wider metaphase plates (iv and vi, black bar indicating width), and misaligned chromosomes (vii–viii, red arrowheads) ($n = 3$ animals/genotype). Scale bars: 10 μm (A i–ii); 2 μm (A iii–viii). (B) Proportion of metaphase spermatocytes exhibiting normal (N), dispersed (Dis), and misaligned chromosomes (Mis) in *Tube1*<sup>Flox/Flox</sup> and *Tube1*<sup>GCKO/GCKO</sup> mice ($n = 3$ animals/genotype, 1 technical replicate; unpaired t-test(s), $P = 0.0016$ (N), $P = 0.0039$ (Dis), $P = 0.0043$ (Mis), **). (C) Binucleated spermatid presence in testes of *Tube1*<sup>GCKO/GCKO</sup> mice shown by PAS-stained testis (black arrowhead) ($n = 3$ animals/genotype). Scale bars: 10 μm. (D–H) Isolated *Tube1*<sup>Flox/Flox</sup> and *Tube1*<sup>GCKO/GCKO</sup> metaphase spermatocytes immunolabeled to visualize centrin (magenta) and β-tubulin (green) and co-stained with DAPI to visualize DNA (blue). Centrin-containing and centrin-less spindle poles indicated by white and yellow arrowheads, respectively. Images are deconvolved and represent 3D maximum intensity projections ($n = 6$ *Flox* and $n = 5$ *GCKO* animals). Scale bars: 2 μm. (I) Proportion of metaphase spermatocytes in *Tube1*<sup>Flox/Flox</sup> and *Tube1*<sup>GCKO/GCKO</sup> mice with abnormal spindle morphology ($n = 6$ *Flox* and $n = 5$ *GCKO* animals, 9–11 *Flox* and 10–13 *GCKO* cells examined/animal; Mann–Whitney U test, $P = 0.0022$, **). (J) Distribution of abnormal metaphase spindle morphologies in *Tube1*<sup>Flox/Flox</sup> and *Tube1*<sup>GCKO/GCKO</sup> mice ($n = 6$ *Flox* and $n = 5$ *GCKO* animals). (K) Pyknotic metaphase and anaphase spermatocytes in *Tube1*<sup>GCKO/GCKO</sup> mice shown by PAS-stained testis (ii, black arrowheads) ($n = 3$ animals/genotype). Scale bars: 20 μm. Data information: For each panel, results are from one experiment. In (B, I), data are presented as mean ± SD. Source data are available online for this figure.

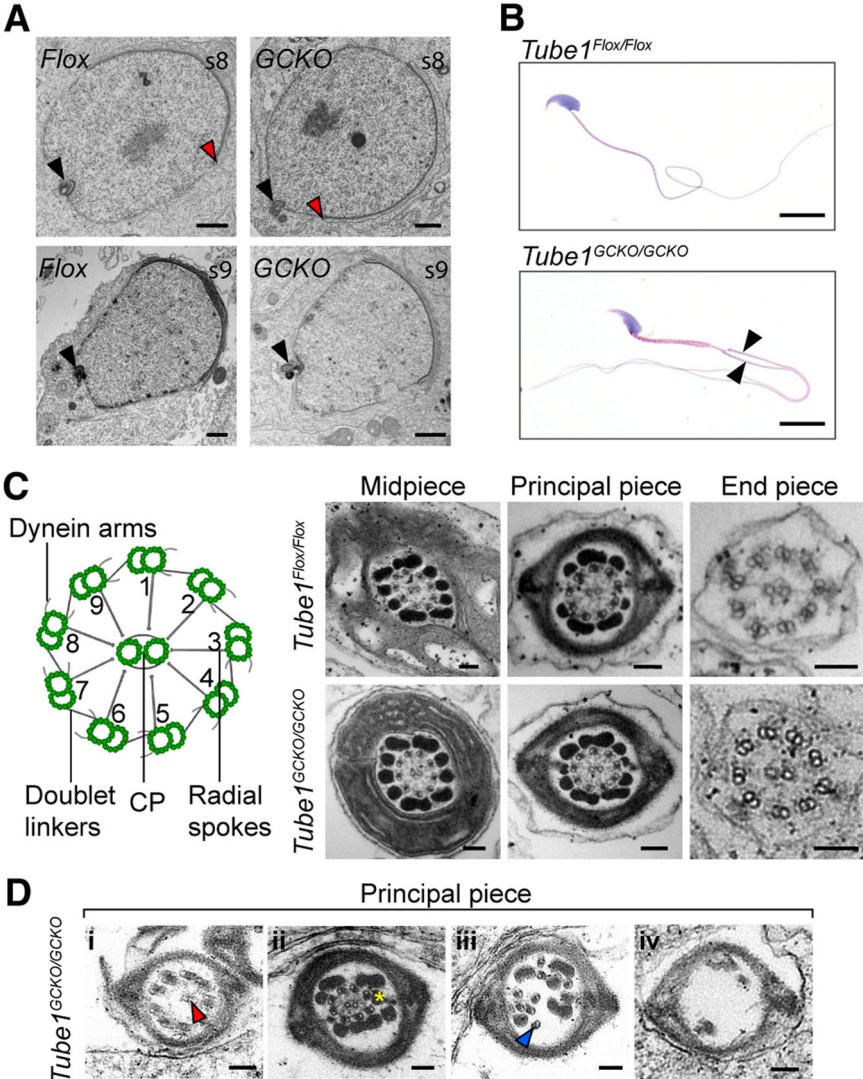

**Figure 3. The absence of TUBE1 compromises basal body function in spermiogenesis.**

(A) TEM displaying basal body docking to nuclear membrane (black arrowheads) in *Tube1*$^{Flox/Flox}$ (*Flox*) and *Tube1*$^{GCKO/GCKO}$ (*GCKO*) testis sections during spermiogenesis. S8 = spermiogenesis step 8; s9 = spermiogenesis step 9. Red arrowheads indicate distal acrosome margin, which is in closer proximity to the basal body docking site in *Tube1*$^{GCKO/GCKO}$. Scale bars: 100 nm. (B) Hematoxylin and eosin stained epididymal sperm. Black arrowheads show two tails emanating from one *Tube1*$^{GCKO/GCKO}$ mouse sperm head. Scale bars: 10 μm. (C) Schematic (left) representing canonical motile axoneme structure and testis cross sections of spermatid axonemes (right), arranged by tail position. CP = central pair. Scale bars: 100 nm. (D) Testis cross sections of the principal piece in *Tube1*$^{GCKO/GCKO}$ mouse sperm with various abnormalities: (i) mispositioned CP (red arrowhead), (ii) missing axonemal doublets (yellow asterisk), (iii) doublets with loss of radial organisation (blue arrowhead), (iv) loss of axoneme and outer dense fiber structure. Scale bars: 100 nm. Data information: For each panel, results are from one experiment. *n* = 3 animals/genotype examined. Source data are available online for this figure.

direct role for TUBE1 in manchette structure and function, testis sections were stained for α-tubulin to visualize manchette microtubules (Fig. 4A). For *Tube1*$^{Flox/Flox}$ and *Tube1*$^{GCKO/GCKO}$ testis sections, manchettes formed as expected, however, and consistent with the abnormal sperm head shapes shown in Fig. 1I, by step 13 of spermatid development (Stage I seminiferous tubules) it was clear that the absence of TUBE1 led to delayed perinuclear ring and manchette migration towards the basal body (Fig. 4A, red arrowheads). Despite this, constriction of the perinuclear ring, and thus of manchette width, continued (as illustrated in Fig. 4A), thus deforming the nucleus. The presence of manchettes also persisted for longer durations in the

*Tube1*$^{GCKO/GCKO}$ model, as shown in step 14 spermatids (Stage II–III) (Fig. 4A), thus suggesting TUBE1 is a target for manchette disassembly at the end of the nuclear elongation process. Consistent with several other models where the manchette failed to migrate (Dunleavy et al, 2021; Dunleavy et al, 2017; O'Donnell et al, 2012), this resulted in the formation of hyper-elongated spermatid nuclei in approximately 50.6% of spermatids, wherein the region of nucleus encircled by the manchette was lengthened (Figs. 4B and EV4). This phenotype is concordant with a failure of manchette-nuclear linker disassembly, which due to the continued growth of manchette microtubules from their plus-ends (localized to the stalled perinuclear ring), resulted in

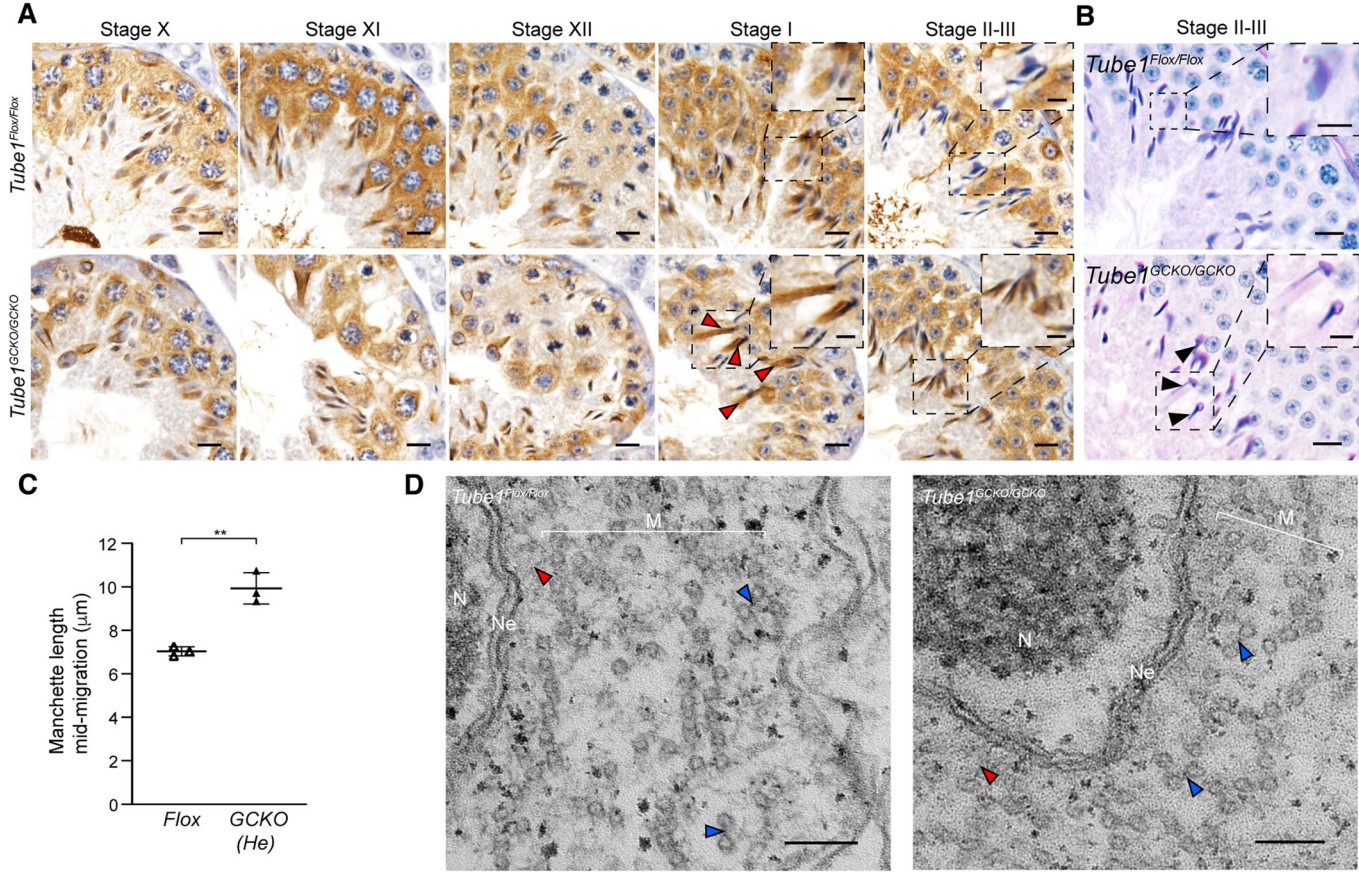

**Figure 4. TUBE1 is required for manchette function.**

(A) Immunohistochemical staining of α-tubulin in Stages X to II–III of spermatogenesis to show manchette progression in the presence and absence of TUBE1. Red arrowheads indicate spermatids in which manchette migration is delayed. Scale bar: 10 μm; 5 μm inset. (B) PAS-stained testis sections, highlighting 'knobby' nuclei (black arrowheads) in the *Tube1GCKO/GCKO* testis. Scale bar: 10 μm; 5 μm inset. (C) Average manchette length (μm) during mid-migration in normal spermatids of *Tube1Flox/Flox* (*Flox*) mice and hyper-elongated (He) spermatids of *Tube1GCKO/GCKO* (*GCKO*) mice (n = 3 animals/genotype, 9 *Flox* and 7–10 *GCKO* cells examined/animal; unpaired t-test, **P = 0.0026; data are presented as mean ± SD). (D) TEM testis sections of elongating spermatids in transverse view. N = nucleus; Ne = nuclear envelope; M = manchette. Blue arrowheads show links between manchette microtubules, red arrowheads show links between microtubules and the nuclear envelope. Scale bars: 100 nm. Data information: For each panel, results are from one experiment. n = 3 animals/genotype examined. Source data are available online for this figure.

the progressive stretching of the spermatid nucleus. Concurrently, manchette microtubules were significantly longer in spermatids with hyper-elongated nuclei (Fig. 4C), suggesting a reduction in minus-end pruning (length regulation) in *Tube1GCKO/GCKO* spermatids. No overt differences in acetylated tubulin immunolabeling were observed between isolated spermatids from *Tube1Flox/Flox* and *Tube1GCKO/GCKO* testes (Fig. EV5), suggesting that while microtubules were longer, there was no change in the proportion of long-lived microtubules at the manchette.

In order to explore the role of TUBE1 in the formation and function of microtubule links between the inner microtubules of the manchette and the nuclear membrane, we used TEM to examine elongating spermatids in the testis. Accompanying the data above showing α-tubulin-labeled manchettes encircling spermatid nuclei in the testis (Fig. 4A), we found links present between the nucleus and manchette in the absence of TUBE1 (Fig. 4D, red arrowheads). In addition, we noted links between manchette microtubules were present in *Tube1GCKO/GCKO* testis sections (Fig. 4D, blue arrowheads). This aside, the failure of

manchette migration strongly points to TUBE1 as a facilitator of manchette-nuclear linker dynamics during the progressive movement of the manchette down the spermatid nuclear membrane and thus, nuclear remodeling. Our data also supports a role for TUBE1 as a target of the microtubule pruning that occurs during nuclear remodeling to maintain manchette length, and subsequently for manchette disassembly.

## TUBE1 loss leads to KATNAL1, KATNAL2, and KATNB1 accumulation during manchette remodeling

We have previously shown the migration and disassembly of the manchette requires KATNAL2, a member of the katanin family of microtubule severing enzymes, and that TUBE1 and KATNAL2 interact at the manchette and pericentriolar region (Dunleavy et al, 2017). In contrast to other katanin A-subunits, KATNAL2 does not sever α-/β-tubulin microtubules when over-expressed in cell lines (Cheung et al, 2016; Dunleavy et al, 2017; Ververis et al, 2016) suggesting a different mechanism of action. TUBE1 being a target

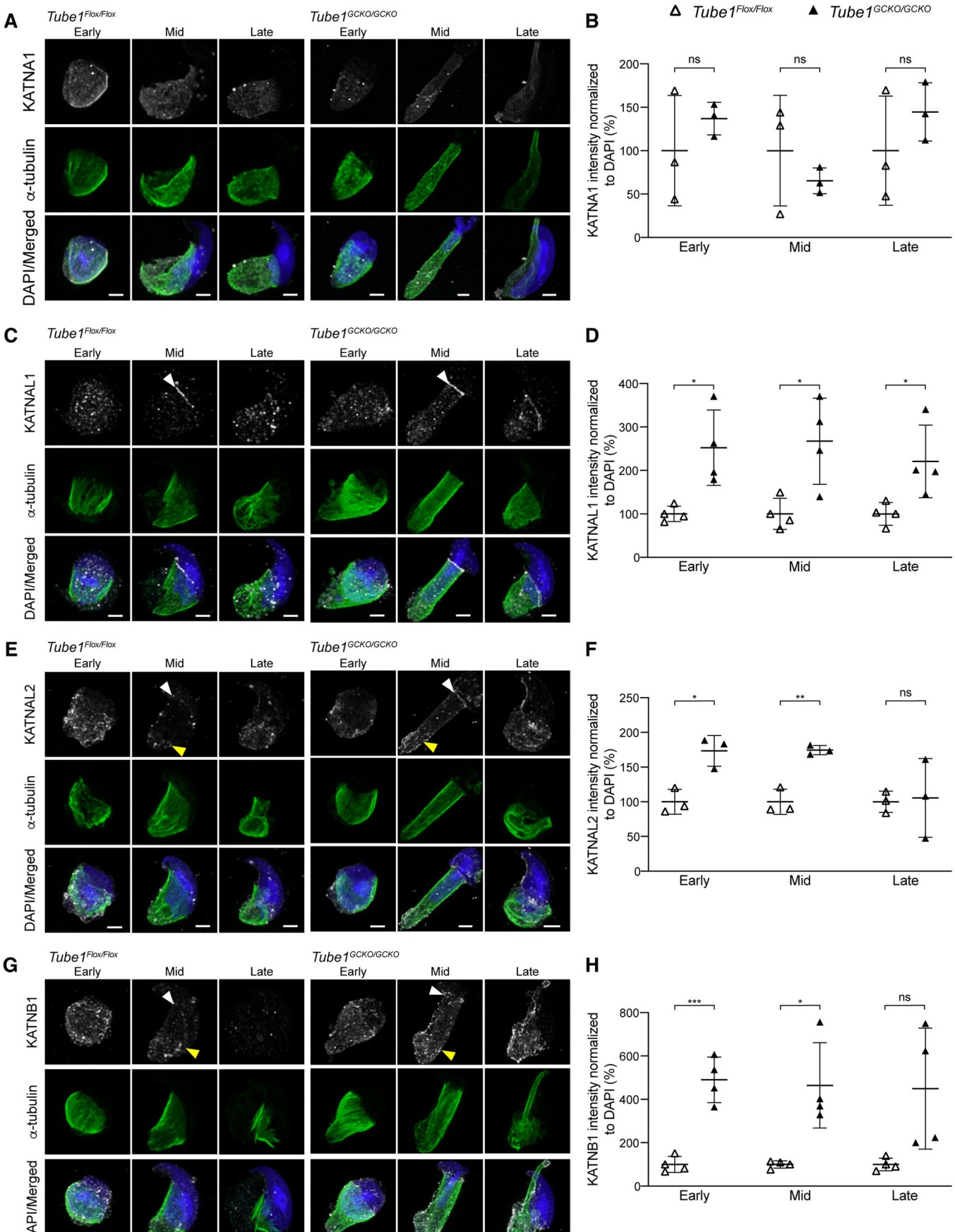

**Figure 5.   TUBE1 loss results in the abnormal accumulation of the katanin microtubule severing enzymes KATNAL1, KATNAL2, and KATNB1.**

(A) Deconvolved Z-stack maximum intensity projections of entire isolated *Tube1*$^{Flox/Flox}$ and *Tube1*$^{GCKO/GCKO}$ elongating spermatids in early, mid, and late manchette migration were immunolabeled to visualize α-tubulin (green) and KATNA1 (gray), then stained with DAPI to visualize the nucleus (blue). (B) Sum katanin subunit intensity values normalized to DAPI (normalized as percentage control group (*Tube1*$^{Flox/Flox}$) to avoid batch staining effects for KATNA1. $n = 3$ animals/genotype, 3 cells analyzed/animal; unpaired t-test(s), $P =$ ns (non-significant; Early, Mid, Late). (C) Deconvolved Z-stack maximum intensity projections of entire isolated *Tube1*$^{Flox/Flox}$ and *Tube1*$^{GCKO/GCKO}$ elongating spermatids in early, mid, and late manchette migration were immunolabeled to visualize α-tubulin (green) and KATNAL1 (gray), then stained with DAPI to visualize the nucleus (blue). (D) Sum katanin subunit intensity values normalized to DAPI (normalized as percentage control group (*Tube1*$^{Flox/Flox}$) to avoid batch staining effects for KATNAL1. $n = 4$ mice/genotype, 3–10 *Tube1*$^{Flox/Flox}$ and 3–7 *Tube1*$^{GCKO/GCKO}$ cells analyzed/animal; unpaired t-test(s), $P = 0.0366$ (Early, Welch's correction), $P = 0.0192$ (Mid), $P = 0.0331$ (Late), *. (E) Deconvolved Z-stack maximum intensity projections of entire isolated *Tube1*$^{Flox/Flox}$ and *Tube1*$^{GCKO/GCKO}$ elongating spermatids in early, mid, and late manchette migration were immunolabeled to visualize α-tubulin (green) and KATNAL2 (gray), then stained with DAPI to visualize the nucleus (blue). (F) Sum katanin subunit intensity values normalized to DAPI (normalized as percentage control group (*Tube1*$^{Flox/Flox}$) to avoid batch staining effects for KATNAL2. $n = 3$ animals/genotype, 2–3 *Tube1*$^{Flox/Flox}$ and 3 *Tube1*$^{GCKO/GCKO}$ cells analyzed/animal; unpaired t-test(s), $P = 0.0111$ (Early), *, $P = 0.0026$ (Mid), **, $P =$ ns (non-significant; Late). (G) Deconvolved Z-stack maximum intensity projections of entire isolated *Tube1*$^{Flox/Flox}$ and *Tube1*$^{GCKO/GCKO}$ elongating spermatids in early, mid, and late manchette migration were immunolabeled to visualize α-tubulin (green) and KATNB1 (gray), then stained with DAPI to visualize the nucleus (blue). (H) Sum katanin subunit intensity values normalized to DAPI (normalized as percentage control group (*Tube1*$^{Flox/Flox}$) to avoid batch staining effects for KATNB1. $n = 4$ mice/genotype, 3 *Tube1*$^{Flox/Flox}$ and 3–5 *Tube1*$^{GCKO/GCKO}$ cells analyzed/animal; unpaired t-test(s), $P = 0.0004$ (Early), ***, $P = 0.0337$ (Mid, Welch's correction), *, $P =$ ns (non-significant; Late, Welch's correction). Data information: Scale bars: 2 μm (A, C, E, G). White arrowheads indicate perinuclear ring localization and yellow arrowheads indicate protein localization at the distal end of the manchette (C, E, G). For each panel, results are from one experiment. In (B, D, F, H), data are presented as mean ± SD). Source data are available online for this figure.

of KATNAL2 severing is supported by the phenocopying of the hyper-elongated and constricted manchette defects in *Katnal2* null (Dunleavy et al, 2017) and *Tube1*$^{GCKO/GCKO}$ mice (Fig. 4). Data presented in this paper, however, reveal the model is more complex than TUBE1 being the single point of linker patency. If this were true, the loss of TUBE1 would lead to the detachment of the manchette from the nuclear membrane.

To explore the consequences of the absence of TUBE1 on katanin subunit recruitment, *Tube1*$^{Flox/Flox}$ and *Tube1*$^{GCKO/GCKO}$ elongating spermatids were purified and immunostained for each of the three katanin severing A-subunits—KATNA1, KATNAL1, and KATNAL2, and the regulatory B-subunit, KATNB1 (Fig. 5). Following image acquisition and deconvolution, we calculated the sum fluorescence intensity normalized to DAPI in each cell, and normalized this as a percentage of the control population stained and imaged in parallel. KATNA1 localized along the manchette in early (step 8–10), mid (step 11–12), and late (step 13–14) elongating spermatids. There was no difference in KATNA1 fluorescence intensity between *Tube1*$^{Flox/Flox}$ and *Tube1*$^{GCKO/GCKO}$ populations (Fig. 5A,B). KATNAL1 was localized along the manchette, most predominantly at the perinuclear ring (Fig. 5C, white arrowheads). Strikingly, *Tube1*$^{GCKO/GCKO}$ mice exhibited an overall 2.46-fold increase in KATNAL1 fluorescence intensity in elongating spermatids compared to *Tube1*$^{Flox/Flox}$ mice (Fig. 5C,D). KATNAL2 also localized along the manchette, to the perinuclear ring (white arrowheads), but more strongly to caudal microtubules of the manchette (Fig. 5E, yellow arrowheads). Moreover, KATNAL2 fluorescence intensity was 1.51-fold higher in *Tube1*$^{GCKO/GCKO}$ mouse spermatids compared to controls (Fig. 5E,F), and was significantly enriched specifically in early and mid-manchette migration populations. In accordance, the regulatory subunit KATNB1 displayed similar localization to KATNAL1 and KATNAL2 and demonstrated a 4.68-fold increase in fluorescence intensity in spermatids from *Tube1*$^{GCKO/GCKO}$ mice (Fig. 5G,H). The over-accumulation of KATNAL1, KATNAL2, and KATNB1 at the manchette and absence of manchette migration and delayed dissolution in our *Tube1*$^{GCKO/GCKO}$ model suggests that TUBE1 is required for the initiation of KATNAL1 and KATNAL2-mediated severing or is a target of KATNAL1 and KATNAL2 mediated severing. In contrast, KATNA1, a katanin that is not required for

manchette migration (Dunleavy et al, 2023), was not affected by the loss of TUBE1.

## Discussion

Like α-, β-, and γ-tubulin, in species where TUBE1 is present it appears to be vital for cell integrity. Herein, we demonstrate that TUBE1 is an essential requirement for male germ cell development and fertility. We establish essential roles for TUBE1 in multiple aspects of male meiosis, including number and positioning of spindle poles, metaphase spindle structure, and cytokinesis to form haploid spermatids. Furthermore, we show that TUBE1 is required for basal body positioning and the formation of a functional and stable axoneme. In contrast to *C. reinhardtii* and *P. tetraurelia* (Dupuis-Williams et al, 2002; Dutcher et al, 2002), in mammalian male germ cells TUBE1 does not appear to be required for the assembly of the characteristic $9 + 2$ axoneme structure within developing mammalian sperm. Caution should, however, be taken in interpreting this data as the possibility remains that a small number of centrioles generated prior to cre-excision may have been inherited throughout spermatogenesis. This aside, the failure to manifest any form of motility with the addition of cell permeable ATP indicates, that key proteins required for motility were missing from sperm generated in *Tube1* null males. Thus, TUBE1 is required for the formation of motile sperm tails. We also demonstrate that TUBE1 plays a critical role in manchette function and ultimately sperm head shaping. Our data strongly suggests that TUBE1 is a target of katanin-mediated remodeling between the manchette and nuclear membrane, however, not for their recruitment. Rather, there is an additional, yet to be unidentified protein(s) required for KATNAL1 and KATNAL2 recruitment to the microtubule-nuclear, and presumably perinuclear ring-nuclear linkers and that this complex of proteins is required for severing of manchette links to allow manchette migration. Finally, we provide data that KATNAL1 and KATNAL2, in complex with KATNB1, each sever distinct elements within the complex microtubule manchette array.

The notable increase in meiotic abnormalities identified in *Tube1* null spermatocytes reveals TUBE1 as a key component of

mammalian male meiosis, and complements previous observations of its requirement for mitosis and centriole stability in cultured somatic cells (Wang et al, 2017). In contrast to mammalian somatic cells, where the absence of TUBE1 reliably led to centriole disintegration prior to mitosis, and supernumeary de novo cenriole formation following mitosis (Wang et al, 2017), mammalian male meiosis was largely centriolar in the absence of TUBE1, except in instances of unfocused and centrin-less spindle poles. Similar to cultured somatic cells, observations of multipolar meiotic spindles with supernumeary centrin foci may suggest an underlying disorder in centriole structure (Wang et al, 2017). Alternatively, super-numerary centrioles may be the consequence of failed cell division. Further, the high proportion of unipolar spindles observed, notably with >2 centrin foci in close proximity, suggest a role for TUBE1 in centrosome separation, or in maintaining the integrity of opposing spindle poles. Collectively these data support a conserved role for TUBE1 in centriole stability between germ and cultured somatic mammalian cells. Future studies will inform the function of TUBE1 in mammalian somatic cells in vivo and enable a comparison between loss of function models.

Herein we identify a new role for TUBE1 in cytokinesis in spermatocytes. This aligns with previous localization of TUBE1 to the midbody during mitosis (Chang and Stearns, 2000) and supports that TUBE1 functions alongside the microtubule severing machinery in spermatogenesis. In accordance, binucleated sperma-tids were seen in KATNAL1, KATNB1 and spastin loss-of-function (LOF) models (Cheers et al, 2023; Dunleavy et al, 2023; Dunleavy et al, 2021).

The formation of axonemes in the mouse spermatids of *Tube1*[GCKO/GCKO] mice is notably different from that seen following TUBE1 loss in unicellular organisms. In *C. reinhardtii* (Dutcher et al, 2002) and *P. tetraurelia* (Dupuis-Williams et al, 2002), TUBE1 was required for axoneme formation, whereas in mouse male germ cells, an axoneme-like structure was formed but it had little capacity for motility. Our data also suggests that TUBE1 is required to maintain axoneme structural integrity in sperm tails. Whether this is a de novo function TUBE1 plays within the basal body or is secondary to defects inherited from spermatocyte centrioles remains to be tested. The mispositioning of the basal body on the nuclear membrane also supports a role for TUBE1 in nuclear docking prior to axoneme extension.

Independent of what appear to be multiple roles for TUBE1 in centrioles and centriole-derived structures, our data reveal that it is required for manchette microtubule migration, length, and timely disassembly during spermiogenesis. However, the precise model of how TUBE1 associates with microtubules within the manchette, and more broadly, remains unknown. In silico analyses suggest that TUBE1 can contact the plus-end of β-tubulin and show that TUBE1 shares considerable homology to one of the lateral surfaces found in α-/β-tubulin, such that it may be capable of making lateral interactions to microtubule filaments (Inclán and Nogales, 2001). Thus, TUBE1 may be incorporated into manchette microtubule filaments, or adjacent to the microtubules themselves. Given that manchettes form in the absence of TUBE1, our data is most consistent with the latter possibility, and moreover suggests that TUBE1 may be an essential accessory for complex microtubule structures that facilitate regulatory processes. In support, the abnormal manchette phenotypes in *Tube1*[GCKO/GCKO] mice were consistent with an absence in microtubule severing activity,

including the failure of manchette-nuclear linker dissolution and the over-accumulation of katanin proteins, and were consistent with our previous data showing an interaction between TUBE1 and KATNAL2 (Dunleavy et al, 2017). This supports the hypothesis that TUBE1 is a target of microtubule severing, however, as indicated above, there must be at least one additional as yet unknown protein involved this complex. As highlighted by the reduction of *Tubd1* mRNA in purified round spermatids, examin-ing TUBD1 function at the manchette should be explored in future studies.

In addition, the over-accumulation of KATNB1 to the manchette in the absence of TUBE1 suggests that KATNAL1 and KATNAL2 severing complexes require both an A- and B-subunit to execute manchette function (Dunleavy et al, 2023; Dunleavy et al, 2017). The differential localization of KATNAL1 and KATNAL2 at the manchette herein, together with our recent data showing that KATNAL1 and KATNAL2 act non-redundantly at the manchette (Dunleavy et al, 2023; Dunleavy et al, 2017), supports a model where KATNAL1 and KATNAL2 have neo-functionalized, from a single katanin A-subunit in lower order species, to distinct paralogs that target different microtubule populations within the manchette. Specifically, our data strongly suggest KATNAL1-KATNB1 com-plexes principally remodel microtubules near, or at the perinuclear ring, whereas KATNAL2-KATNB1 complexes target microtubules interfacing the nuclear membrane within the manchette bulk, and during manchette disassembly.

Combining these new insights, we put forward an updated hypothetical model of manchette regulation, which occurs in four cycling phases: Bind, Cut, Migrate + Constrict and Relink (Fig. 6). This model proposes that the links between (1) the perinuclear ring and the nucleus and (2) the manchette microtubules and the nucleus are progressively remodeled by (1) KATNAL1-KATNB1 and (2) KATNAL2-KATNB1 complexes, respectively, in a TUBE1-dependent manner, to enable the manchette to rachet down the nucleus. While migration is occurring, the perinuclear ring constricts via an unknown mechanism to shape the nucleus (Lehti and Sironen, 2016). Equally, data presented here, and in the previous katanin loss-of-function models (Dunleavy et al, 2023; Dunleavy et al, 2021; Dunleavy et al, 2017) supports the continuous incorporation of tubulin subunits at the dynamic plus ends of individual microtubules within the skirt of the manchette. This results in the 'pushing' down (caudally) of the microtubule body and contributes to the elongation of the spermatid nucleus. Under normal situations we hypothesize that such plus-end growth is at least partially counterbalanced by the action of katanin complexes systematically severing the links between the manchette (peri-nuclear ring and microtubule body) and minus-end. In addition, and in order to prevent the manchette detaching from the nuclear membrane we propose that perinuclear ring-nuclear and manchette-nuclear links must be remodeled in a coordinated and rolling cycle of severing, migration, and reforming. Finally, towards the end of spermatogenesis KATNAL2-KATNB1 complexes depolymerize microtubules within and at the manchette minus-end (Fig. 6). In the absence of TUBE1, katanin severing complexes continue to be recruited to the manchette by an unknown protein(s) but are unable to execute severing and thus over-accumulate. We predict that the absence of TUBE1 results in nuclear membrane linkers remaining intact and microtubule pruning is diminished, ultimately leading to failure in manchette

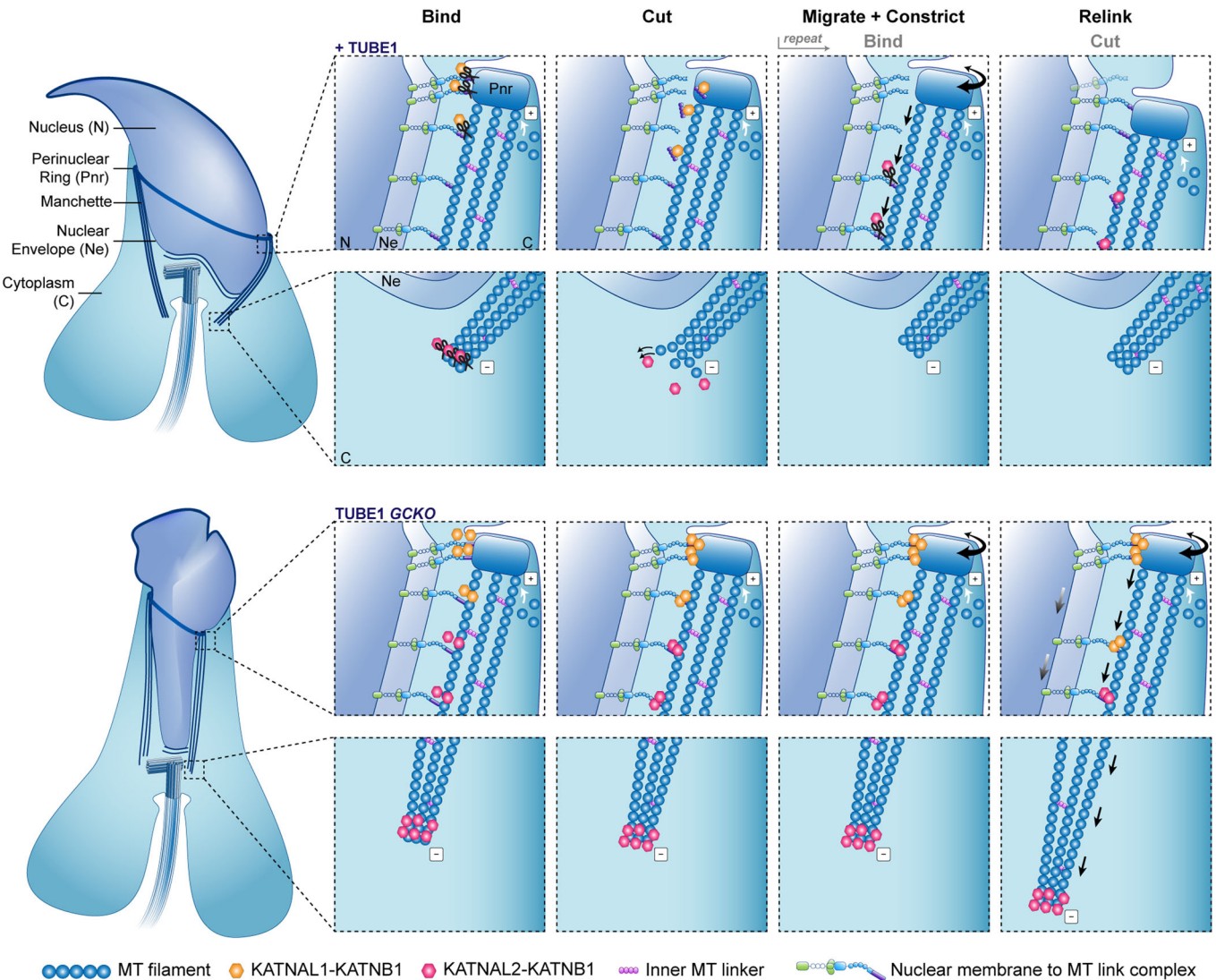

**Figure 6. Hypothesis of the mechanism of manchette migration in the mouse.**

Schematic depicting the proposed cyclical process of katanin-mediated manchette migration in mouse elongating spermatids in the presence (+TUBE1) and the absence (TUBE1 *GCKO*) of TUBE1. When TUBE1 is present, we believe KATNAL1-KATNB1 and KATNAL2-KATNB1 complexes are recruited to the manchette microtubules (MTs) and perinuclear ring linkers by an as yet unidentified protein (Bind phase). We propose KATNAL1-KATNB1 and KATNAL2-KATNB1 complexes sever the microtubule-nuclear and perinuclear ring-nuclear links (Cut phase). This facilitates ratcheting of the manchette down the nucleus (straight black arrows) while the perinuclear ring concurrently constricts in a TUBE1-independent manner (double-sided curved arrow) (Migrate + Constrict phase). In a rolling manner, we suggest links between the nucleus and inner manchette microtubules and perinuclear ring are reattached, thus preventing manchette detachment (Relink phase). In parallel, new microtubule dimers are incorporated at the plus-end of the microtubule skirt and this is counterbalanced by KATNAL1-KATNB1 and KATNAL2-KATNB1 complexes regulating manchette length by pruning microtubules along the manchette body and/or minus-end. When TUBE1 is absent, katanin complexes continue to be recruited to the manchette, but are unable to complete the Cut phase and consequently accumulate. The manchette migration and pruning thus cannot occur. Perinuclear ring constriction, however, continues, and tubulins continue to be incorporated into the plus ends of the microtubule skirt (white arrows). This loss of co-ordination between parallel processes within the manchette results in stretching of the nucleus and ultimately hyper-elongated manchettes and thus nuclei (gradient arrow). Signs '+' and '−' indicate microtubule plus- and minus-ends, respectively.

migration and hyper-extension of the manchette microtubules. Despite this, perinuclear ring constriction proceeds, and tubulin subunits continue to be incorporated at the plus-ends of the skirt of the manchette microtubule in the absence of TUBE1, collectively resulting in the distal half of the nucleus becoming hyper-elongated and constricted (Fig. 6).

While the current study provides numerous new elements to support this proposed hypothetical model, several aspects remain

untested. Future studies are needed to directly visualize tubulin dimer incorporation at microtubule plus-end during manchette movement. Definitive experiments are required to demonstrate TUBE1 is a target of katanin severing and the identity of cofactors involved in microtubule-nuclear linker. Importantly, how katanins, and KATNAL2 in particular, targets TUBE1 remains unknown. TUBE1 does not possess an acidic C-terminal tail, the target of traditional KATNA1-KATNB1 (and presumably KATNAL1-

KATNB1) microtubule severing (Zehr et al, 2020). KATNAL1 does, however, possess a N-terminal microtubule interacting and trafficking domain (MIT) (McNally and Roll-Mecak, 2018), and tubulin-MIT interactions have been proposed as the driving factor underlying microtubule depolymerization independent of the tubulin C-terminal tail (Belonogov et al, 2019). However, if and how TUBE1 is incorporated into microtubules is not currently known.

The mechanism underlying KATNAL2 function is yet to be uncovered. Its sequence is notably divergent from KATNA1 and KATNAL1, and while KATNAL2 retains the AAA ATPase domain, it possesses a lissencephaly homology domain (LisH) in place of a MIT domain, suggesting it recognizes and interacts with microtubules differently to KATNA1 and KATNAL1 (Shin et al, 2019). A difference in severing mechanism is supported by the absence of overt KATNAL2 severing of α-/β-tubulin polymers when transfected into somatic cells (Cheung et al, 2016; Dunleavy et al, 2017; Ververis et al, 2016). The ability to sever is however strongly evidenced by the defects in the microtubule germ cell cytoskeleton in the absence of KATNAL2 (Dunleavy et al, 2017).

Collectively, our data establish TUBE1 as an essential regulator of complex microtubule structures in the male germ cell cytoskeleton. While these structures, such as the manchette, midbody, centriole and axoneme form in the absence of TUBE1, their regulation, function and structural integrity is fatally compromised, revealing that TUBE1 is required for germ cell development and male fertility. Furthermore, we reveal the dependence between TUBE1 and katanin proteins and propose a mechanism for the movement of a complex microtubule array, the manchette, and nuclear remodeling.

# Methods

## Animal ethics statement

All animal procedures were approved by the Monash University School of Biological Sciences Animal Experimentation Ethics Committee (Ethics number 14606) and the University of Melbourne Ethics Committee (Ethics number 20640) and were conducted in accordance with Australian National Health and Medical Research Council (NHMRC) Guidelines on Ethics in Animal Experimentation and the Animal Research: Reporting of In Vivo Experiments guidelines.

## Mouse model production

A mouse model with loxP sites flanking ENSMUSE00000098484 (Exon 3) of the Tube1 gene was generated at the Monash Genome Modification Platform using CRISPR/Cas9 standard methods. Guide RNA target sites (5′ end of exon—5′ TACCAGCTGTA-TAATCCTAC 3′; 3′ end of exon—5′ CCGCTAGACTCATACCA-TAG 3′) were identified using the UCSC Genome Browser (https://genome.ucsc.edu/). A Tube1 conditional KO targeting construct was used to generate a single-stranded DNA homology repair template, which amplified a product between 5′/5Phos/AGTC ACAGGCAGAAGGCACTTG 3′ and 5′ AGACCCAGGCATAT CTCAAATGTCA 3. The CRISPR/Cas9 ribonucleoprotein complex and single-stranded DNA homology repair templates were

microinjected into the pronucleus of C57BL/6J zygotes, which were transferred into the uterus of pseudo pregnant C57BL/6J females.

Mice homozygous for this inserted sequence ($Tube1^{Flox/Flox}$) were crossed to C57BL/6J Stra8-Cre transgenic mice using a two-step breeding strategy, to achieve a conditional knockout through Cre-lox recombination. (i) $Tube1^{Flox/Flox}$ females were mated with Stra8-Cre males (Sadate-Ngatchou et al, 2008) to create $Tube1^{WT/Flox;\ Stra8-Cre}$ offspring. (ii) $Tube1^{Flox/Flox}$ females were mated with $Tube1^{WT/Flox;\ Stra8-Cre}$ males to generate germ cell-specific knockout, $Tube1^{Flox/Flox;\ Stra8-Cre}$ ($Tube1^{GCKO/GCKO}$), offspring. With Stra8-Cre expressed between early-spermatogonia and pre-leptotene spermatocytes (Sadate-Ngatchou et al, 2008), this allows us to examine the consequences of Tube1 loss in a pre-meiotic deletion model. The genotypes of these mice were determined using real time PCR and allele-specific probes (Transnetyx, Cordova, TN). Excision of the flanked exon 3 allele was confirmed via qPCR on spermatocyte-enriched germ cell populations. Mice were maintained on a C57BL/6J background. Mice were housed in a pathogen-free facility on a 14-h light/10-h dark cycle maintained at 21 °C, with free access to filtered water and irradiated chow. Male adult mice (>50 days of age) were used in this study, with a minimum of three mice used, unblinded, per genotype and per experiment. The means to make these mice are available upon request.

## Germ cell isolation

Purified germ cell populations of spermatocytes, round spermatids, elongating spermatids, and elongated spermatids were obtained using the STAPUT method detailed in Dunleavy et al (2019b). For immunostaining, cells were either fixed with 4% paraformaldehyde at room temperature for 10 min or settled onto poly-L-lysine-coated coverslips for 15 min at room temperature, and then fixed in 100% methanol at −20 °C for 5 min.

## Quantitative PCR

Expression of Tube1 and Tubd1 mRNA in purified germ cells was performed as detailed in Dunleavy et al (2021). A PCR primer set was designed to span part of exons 2 and 3 of mouse Tube1 (5′-GA CAGTGCGGGAACCAGAT-3′ forward and 5′-AGCTGCTTATAG CGTCGTCA-3′ reverse). A PCR primer set was designed to span part of exons 7 and 8 of Tubd1 (5′-CAGTGCAGACGTAGAG GGATT-3′ forward and 5′-GGCTCGCTGGGTTTTCCATA-3′ reverse), Peptidylprolyl isomerase A (Ppia) (5′-GTCTCCTTCGA GCTGTTT-3′ forward and 5′-ACCCTGGACATGAATCCT-3′ reverse) and Hypoxanthine phosphoribosyl transferase 1 (Hprt1) (5′-AGTCCCAGCGTCGTGATTAG-3′ forward and 5′-TTTCCA AATCCTCGGCATAA-3′ reverse) were used as reference genes.

## Fertility characterization

Male fertility in $Tube1^{GCKO/GCKO}$ male mice was tested and characterized following the guidelines set out in Houston et al (2020). The staging of seminiferous tubule sections and spermatid steps was performed according to Russell et al (1990). For breeding trials, $Tube1^{Flox/Flox}$ and $Tube1^{GCKO/GCKO}$ male mice of 10–12 weeks of age were bred with two wild type (C57BL/6J) female mice each (≥6 weeks of age), with mating confirmed by the presence of copulatory plugs. Testis and epididymal tissue was collected for

histology, alongside epididymal sperm, and was assessed as previously described (Borg et al, 2009). Epididymal sperm head shapes were manually graded and categorized as 'normal', 'narrow', or 'hammer shape' ($n \geq 29$ cells/animal; $n = 3$ animals/genotype). Spermatid shapes were manually graded from Stage IX PAS-stained testis sections ($n = 3$ seminiferous tubules/animal; $n = 3$ animals/genotype). The number of functional sperm produced was calculated by multiplying the values of epididymal sperm content by progressive motility in $Tube1^{GCKO/GCKO}$ mice.

A modified version of the Triton X-100 nuclear solubilization method was used to determine daily sperm production and total epididymal sperm count (Dunleavy et al, 2021). Sperm motility was assessed using a computer-assisted sperm analyzer, as described previously (Gibbs et al, 2011). To test sperm motility response to ATP, CASA was performed before and after the addition of 55 µg/ml Adenosine 5′-triphosphate disodium salt hydrate (A2383, Sigma). Given the low numbers of sperm present in epididymides of $Tube1^{GCKO/GCKO}$ mice a minimum 80 sperm/animal were counted.

Testes and epididymides were fixed in Bouin's fixative as previously described (Dunleavy et al, 2017). PAS alongside hematoxylin counterstaining, was used to assess the histology of tissue sections. Germ cell apoptosis was evaluated by immunohistochemistry staining for cleaved Caspase 3 and 7, and hematoxylin counterstaining, where the number of Caspase-positive cells present, per seminiferous tubule section, per mouse was counted. Statistical analyses were conducted as detailed in Dunleavy et al (2021) and below.

## Antibody staining and immunochemistry

Testis sections and germ cells were processed and stained as previously described (Dunleavy et al, 2021). Primary antibodies included those against α-tubulin (T5168, Sigma-Aldrich, ascites fluid, 1:5000 for immunofluorescence, 1:50,000 for immunohistochemistry), acetylated tubulin (T6793, Sigma, 1:1000), β-tubulin (ab21057, Abcam, 1 µg ml$^{-1}$), γ-tubulin (ab11317, Abcam, 10.9 µg ml$^{-1}$), centrin 20H5 (04-1624, Merck Millipore, 1 µg ml$^{-1}$), cleaved Caspase 3 (9664, Cell Signaling Technology, 0.5 µg ml$^{-1}$), cleaved Caspase 7 (9491, Cell Signaling Technology, 1:500), KATNA1, KATNAL1, and KATNB1 were used as described previously (Dunleavy et al, 2021), KATNAL2 (as described previously; Dunleavy et al, 2017), SYCP3 (ab15093, Abcam, 10 µg ml$^{-1}$). Secondary antibodies were used at a 1:500 dilution and included Alexa Fluor 488 donkey anti-mouse (A-21202, Invitrogen), Alexa Fluor 555 donkey anti-goat (A-21432, Invitrogen), Alexa Fluor 555 donkey anti-rabbit (A-31572, Invitrogen), Alexa Fluor 555 donkey anti-rat (A-78945, Invitrogen), Alexa Fluor 647 donkey anti-mouse (A-31571, Invitrogen), Alexa Fluor 647 donkey anti-goat (A-21447, Invitrogen) and EnVision+ Dual Link System HRP (K4063, Dako).

Immunofluorescent images were captured on a Leica SP8 confocal microscope (Biological Optical Microscopy Platform, The University of Melbourne, Parkville). All fluorescent images were captured using a 63x/1.40 HC PL APO CS2 oil immersion objective. Z-stacks were taken between 0.15 µm and 0.3 µm intervals and 3D maximum intensity projections were generated using Imaris Viewer 9.8.0. Adobe Photoshop was used to uniformly adjust images.

## Image restoration and analysis

Confocal images were deconvolved using Huygens Professional versions 21.10 and 22.04 (Scientific Volume Imaging, The Netherlands, http://svi.nl) with standard express deconvolution parameters. Imaris 9.6.1 was used to create surfaces of antibody-stained channels to provide measurements of sum intensity. Sum katanin intensity values were normalized to DAPI and the control ($Tube1^{Flox/Flox}$) population in each group. Imaris 9.6.1 was also used to measure α-tubulin-labeled manchette lengths in isolated mid-manchette migration spermatids.

## Transmission electron microscopy

To examine seminiferous tubule ultrastructure, testes were fixed, processed and cut for TEM as previously described (Dunleavy et al, 2021). Sections were examined on a JEM-1400 Plus TEM in the Monash Ramaciotti Centre for Cryo-Electron Microscopy (Monash University) and on a FEI Talos L120CI TEM in the Ian Holmes Imaging Centre (The University of Melbourne).

## Statistical analysis

Quantification of germ cell apoptosis was evaluated by counting the number of cleaved Caspase 3 and cleaved Caspase 7 positive cells per tubule in a testis cross section. Statistical analysis was performed using R version 4.1.10 (R Core Team, 2021) and R Studio version 1.4.1717 (RStudio Team, 2020), alongside the packages: lme4 (Bates et al, 2015), glmmTMB (Brooks et al, 2017). A generalized linear model with zero-inflated negative binomial distribution was selected for analyses. This model was deemed most appropriate, based on Akaike information criteria (AIC) estimates.

All other statistical analyses were conducted in GraphPad Prism 9.0, with the name of each test reported in figure legends. Following assessment of normal distribution using a Shapiro–Wilk test, a paired two-tailed t-test analyzed sperm mean velocity before and after treatment with membrane permeable ATP between $Tube1^{Flox/Flox}$ and $Tube1^{GCKO/GCKO}$ populations. For all other analyses between $Tube1^{Flox/Flox}$ and $Tube1^{GCKO/GCKO}$ populations, a Shapiro–Wilk test was used to assess normal distribution and a F-test was used to assess even variance. In the case where data was both normally distributed with even variance, an unpaired two-tailed t-test was used. Where data was normally distributed, with uneven variance, an unpaired two-tailed t-test with Welch's correction was used. Where data was non-normally distributed, a non-parametric Mann–Whitney U test was used. Statistical significance is indicated as $*P < 0.05$, $**P < 0.01$, $***P < 0.001$, and $****P < 0.0001$.

# Data availability

This study includes no data deposited in external repositories. The source data of this paper are collected in the following database record: biostudies:S-SCDT-10_1038-S44319-024-00159-w.

# Peer review information

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

## Acknowledgements

The authors thank Dr. Brendan Houston, Dr. Margot von Kopie and Joseph Nguyen for technical assistance. The authors thank the following platforms for use of their facilities and services: at The University of Melbourne (Parkville), the Biological Optical Imaging Platform, the Ian Holmes Imaging Center, and the Melbourne Histology Platform; and at Monash University (Clayton), the Monash Animal Research Platform, the Monash Histology Platform, and the Monash Ramaciotti Centre for Cryo-Electron Microscopy. GGS is supported by an Australian Government Research Training Program Scholarship. JEMD is supported by a National Health and Medical Research Council Ideas Grant to MKO'B and JEMD (APP1180929) and this project was supported in part by funding from the Australian Research Council (DP160100647). JZ is supported by the National Health and Medical Research Council (APP2002507 and APP2009409), the Sylvia and Charles Viertel Charitable Foundation and the Canadian Institute for Advanced Research (CIFAR) Azrieli Scholarship. The Australian Regenerative Medicine Institute is supported by grants from the State Government of Victoria and the Australian Government.

## Author contributions

**G Gemma Stathatos**: Conceptualization; Formal analysis; Investigation; Visualization; Methodology; Writing—original draft; Project administration; Writing—review and editing. **D Jo Merriner**: Formal analysis; Investigation; Methodology; Writing—review and editing. **Anne E O'Connor**: Formal analysis; Methodology; Project administration; Writing—review and editing. **Jennifer Zenker**: Visualization; Writing—original draft; Writing—review and editing. **Jessica EM Dunleavy**: Conceptualization; Formal analysis; Investigation; Visualization; Methodology; Writing—original draft; Project administration; Writing—review and editing. **Moira K O'Bryan**: Conceptualization; Formal analysis; Supervision; Funding acquisition; Investigation; Visualization; Writing—original draft; Project administration; Writing—review and editing.

Source data underlying figure panels in this paper may have individual authorship assigned. Where available, figure panel/source data authorship is listed in the following database record: biostudies:S-SCDT-10_1038-S44319-024-00159-w.

## Disclosure and competing interests statement

The authors declare no competing interests.

# Expanded View Figures

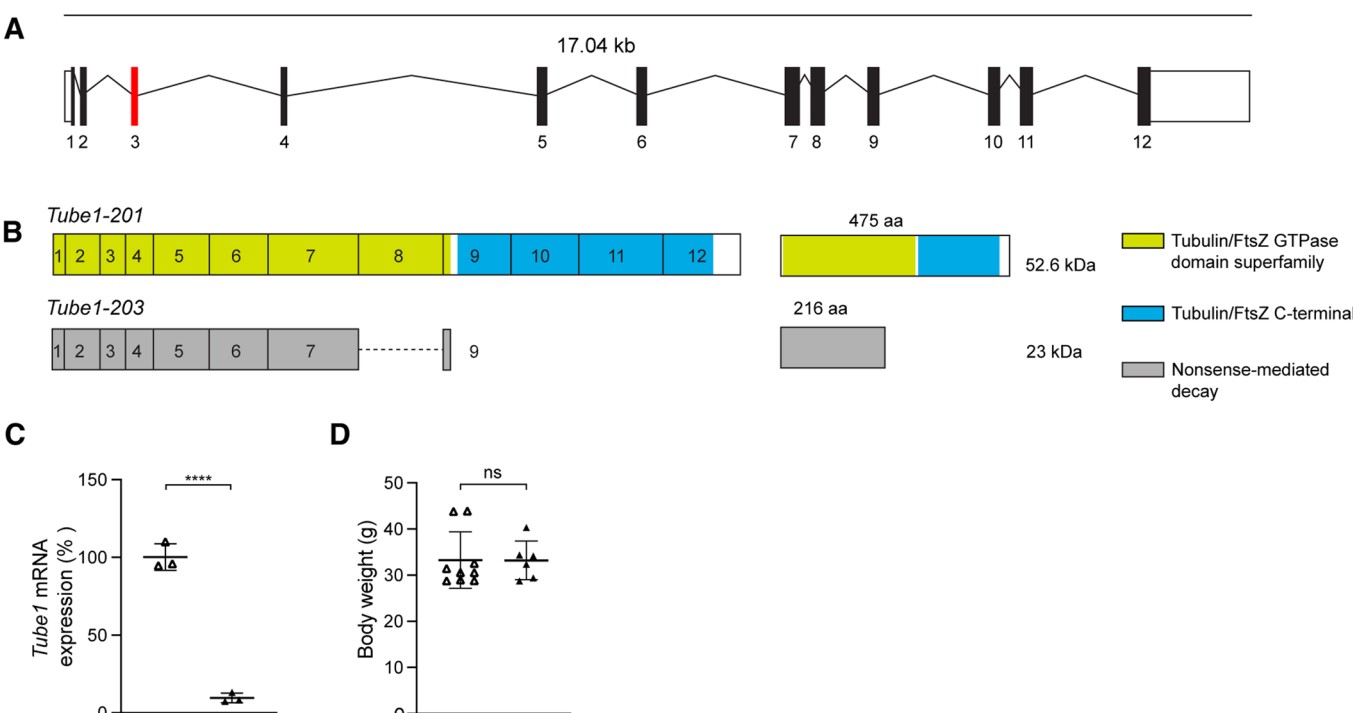

**Figure EV1. Creation of the *Tube1^GCKO/GCKO* mouse model and *Tube1* expression.**

(**A**) The *Tube1* gene consists of 12 exons, with length shown in kilobases (kb). Exons are boxes and introns are lines. Exon 3 targeted for deletion in the male germline highlighted in red. (**B**) Processed transcripts of *Tube1* gene as named in Ensembl (ENSMUSG00000019845). *Tube1-201* is the only predicted protein coding transcript, whereas *Tube1-203* undergoes nonsense-mediated decay (Cunningham et al, 2021). (**C**) *Tube1* mRNA expression in STAPUT purified spermatocytes as percentage of control from *Tube1^Flox/Flox* (*Flox*) and *Tube1^GCKO/GCKO* (*GCKO*) mice, with *Ppia* as a reference gene (n = 3 animals/genotype, 3 technical replicates; unpaired t-test, P < 0.0001, ****). (**D**) Body weight comparison (grams, g) between *Tube1^Flox/Flox* and *Tube1^GCKO/GCKO* male mice (n = 9 *Flox* and n = 6 *GCKO* animals, 1 technical replicate; Mann–Whitney U test, P = ns (non-significant)). Data information: For each data panel, results are from one experiment. In (**C**, **D**), data are presented as mean ± SD). Source data are available online for this figure.

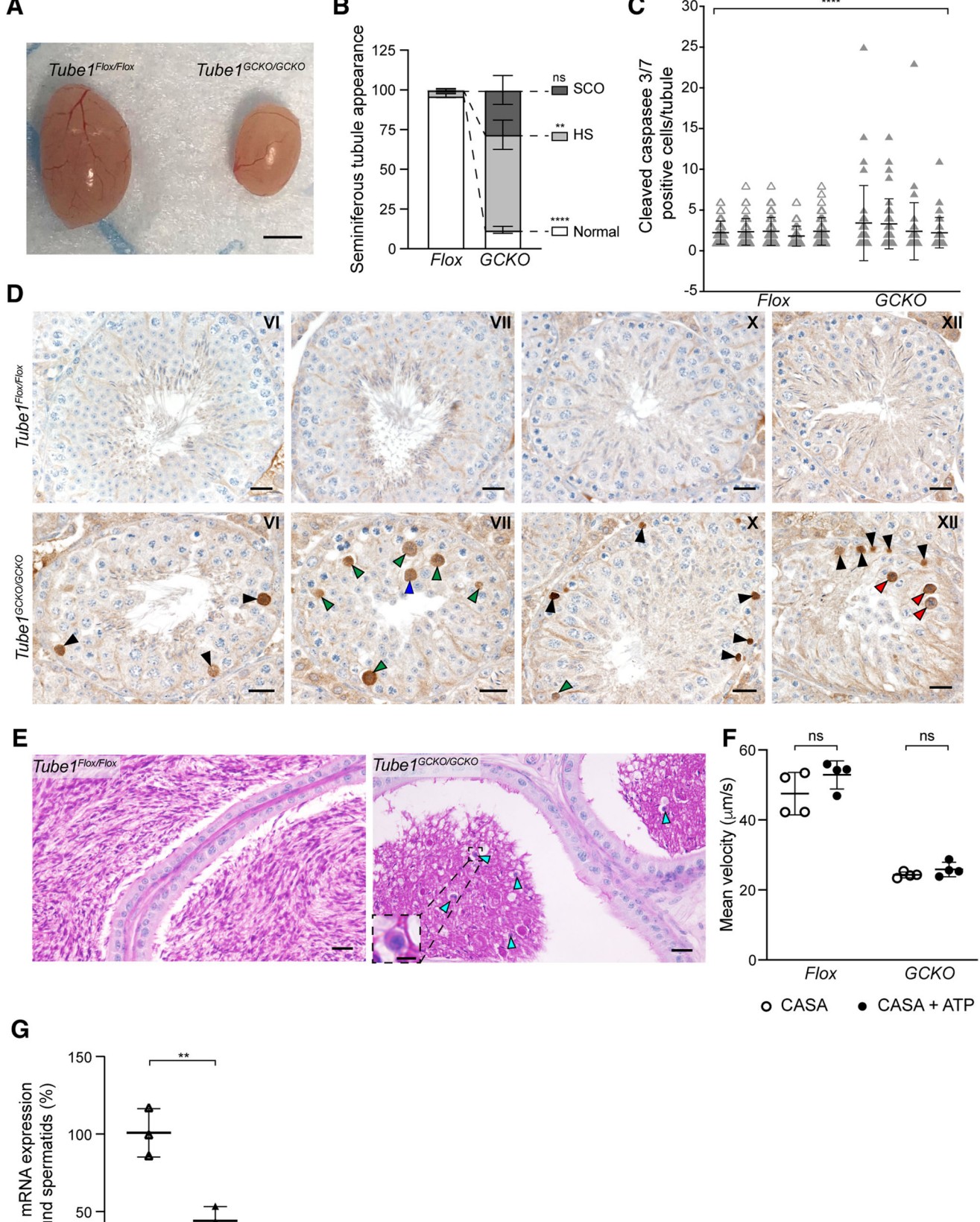

◀ **Figure EV2.  TUBE1 is required for spermatogenesis in the mouse.**

(A) Representative images of a testis from *Tube1^Flox/Flox^* and *Tube1^GCKO/GCKO^* mice. Scale bars: 2 mm. (B) Distribution of normal spermatogenesis, hypospermatogenesis (HS), or Sertoli cell only (SCO) tubules observed within seminiferous tubule cross sections from *Tube1^Flox/Flox^* (*Flox*) and *Tube1^GCKO/GCKO^* (*GCKO*) mice ($n = 3$ animals/genotype, 1 technical replicate; unpaired t-test(s), $P < 0.0001$ (Normal), ****, $P = 0.0080$ (Hypo, Welch's correction), **, and Mann–Whitney U test, $P = $ ns (non-significant; SCO)). (C) Average number of cleaved Caspase 3 and/or 7 positive cells per seminiferous tubule, per mouse from *Tube1^Flox/Flox^* (*Flox*) and *Tube1^GCKO/GCKO^* (*GCKO*) mice ($n = 5$ *Flox* and $n = 4$ *GCKO* animals, 1 technical replicate; generalized linear model with zero-inflated negative binomial distribution, $P < 0.0001$, ****). (D) Staged cross sections of seminiferous tubules stained for cleaved Caspase 3 and 7. Caspase-positive cells identified as spermatogonia (black arrowheads), pachytene spermatocytes (green arrowheads), round spermatid (blue arrowhead) and metaphase spermatocytes (red arrowheads) ($n = 5$ *Flox* and $n = 4$ *GCKO* animals). Scale bars: 20 µm. (E) PAS-stained epididymis sections illustrating germ cell sloughing in *Tube1^GCKO/GCKO^* mice (cyan arrowheads) ($n = 3$ animals/genotype). Scale bars: 20 µm; 5 µm inset. (F) Mean velocity (µm/s) of *Flox* and *GCKO* epididymal sperm measured using CASA, before (CASA) and after (CASA + ATP) the addition of 55 µg/ml ATP ($n = 3$ animals/genotype, 1 technical replicate; paired t-test(s), $P = $ ns (non-significant; *Flox*, *GCKO*)). (G) *Tubd1* mRNA expression in STAPUT purified round spermatids as percentage of control from *Tube1^Flox/Flox^* (*Flox*) and *Tube1^GCKO/GCKO^* (*GCKO*) mice, with *Hprt1* and *Ppia* as reference genes ($n = 3$ animals/genotype, 3 technical replicates; unpaired t-test, $P = 0.0055$, **). Data information: For each data panel, results are from one experiment. In (B, C, F, G), data are presented as mean ± SD). Scale bars: 20 µm; 5 µm inset. Source data are available online for this figure.

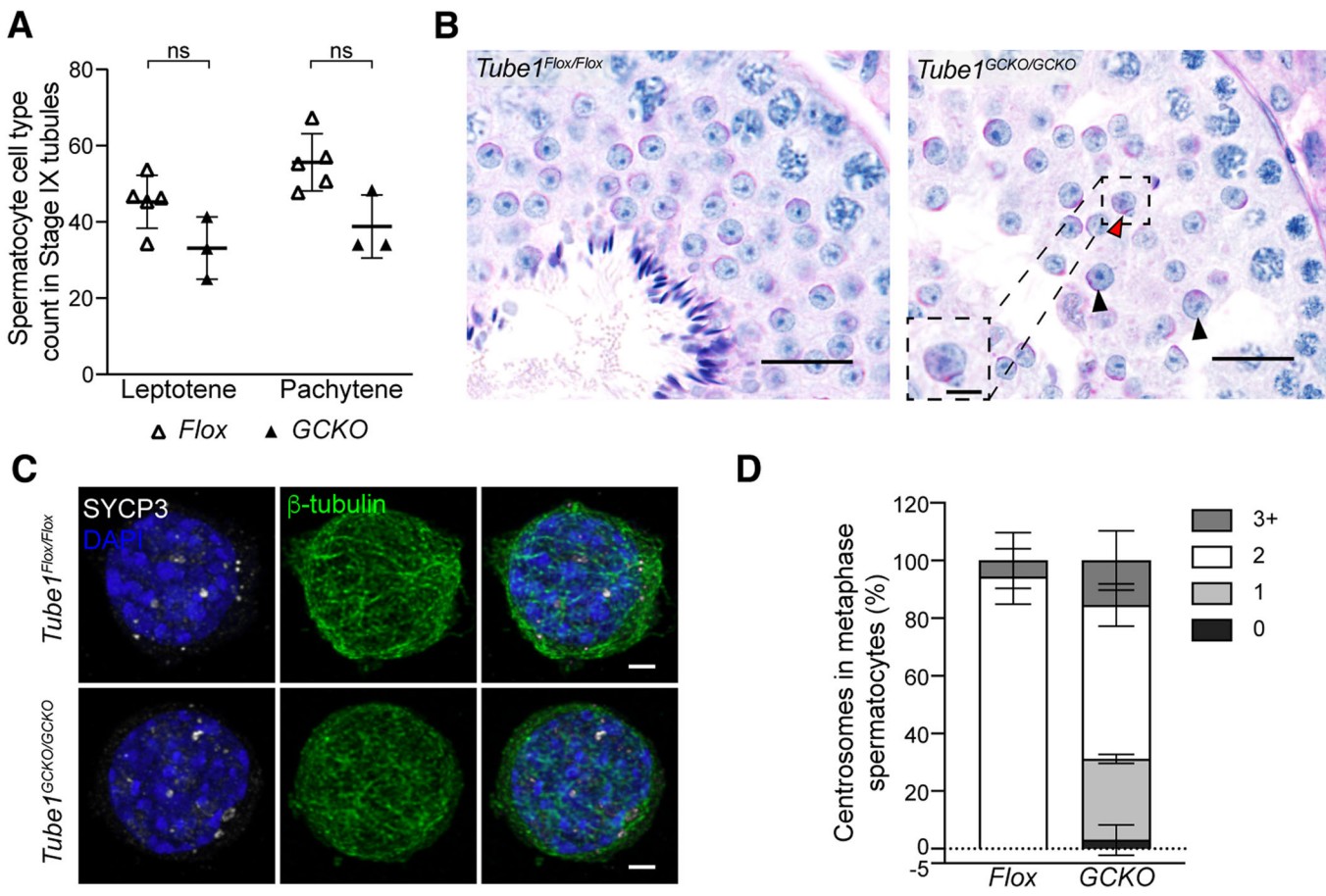

**Figure EV3.    The beginning of meiosis prophase I is not affected by TUBE1 loss-of-function.**

(A) Average number of leptotene and pachytene spermatocytes in Stage IX seminiferous tubules in *Tube1^Flox/Flox^* (*Flox*) and *Tube1^GCKO/GCKO^* (*GCKO*) mice (*n* = 5 *Flox* and *n* = 3 *GCKO* animals, 3 seminiferous tubules examined per animal; unpaired t-test (leptotene), Mann–Whitney U test (pachytene), *P* = ns (non-significant)). Data are presented as mean ± SD. (B) Cross section of seminiferous tubules with abnormally large round spermatid nuclei in *Tube1^GCKO/GCKO^* (black arrowheads) and a binucleated spermatid with one small nucleus and one large nucleus (red arrowheads) (*n* = 3 animals/genotype). Scale bars: 20 μm. (C) Isolated *Tube1^Flox/Flox^* and *Tube1^GCKO/GCKO^* leptotene spermatocytes immunolabeled to visualize synaptonemal complex protein 3 (SYCP3, white) and β-tubulin (green) and co-stained with DAPI to visualize DNA (blue). Images are deconvolved and represent 3D maximum intensity projections (*n* = 3 *Flox* and *n* = 3 *GCKO* animals). Scale bars: 2 μm. (D) Percentage distribution of centrosome number in metaphase spermatocytes from *Tube1^Flox/Flox^* and *Tube1^GCKO/GCKO^* mice (*n* = 3 animals/genotype, 6–9 *Tube1^Flox/Flox^* and 10–11 *Tube1^GCKO/GCKO^* cells analyzed/animal). Data are presented as mean ± SD. Source data are available online for this figure.

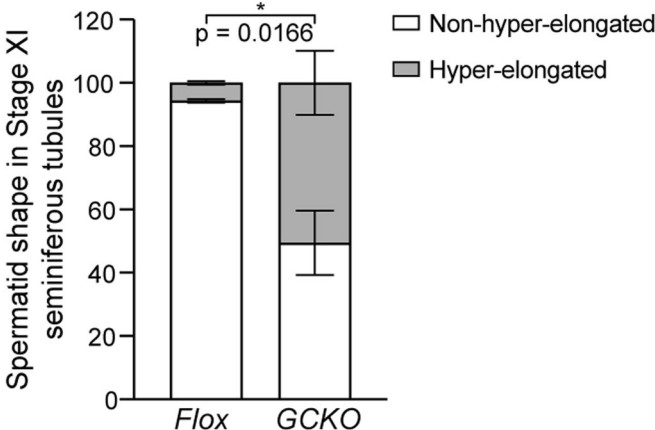

**Figure EV4. Increase in hyper-elongated spermatid nuclei in the absence of TUBE1.**

Distribution of non-hyper-elongated and hyper-elongated spermatid shapes observed within seminiferous tubule cross sections from *Tube1^Flox/Flox^* (*Flox*) and *Tube1^GCKO/GCKO^* (*GCKO*) mice ($n = 3$ animals/genotype, 1 technical replicate; unpaired t-test, $P = 0.0166$ (Welch's correction), \*). Results are from one experiment. Source data are available online for this figure.

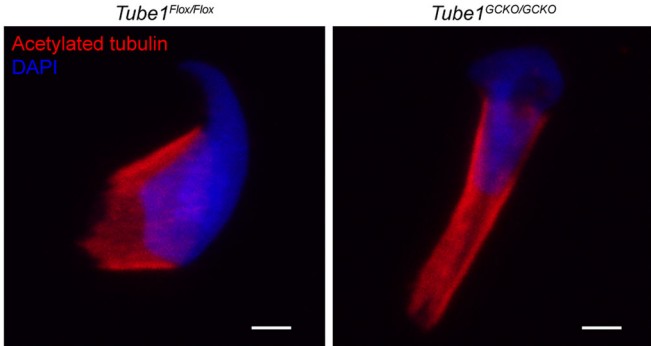

**Figure EV5. Manchette acetylated tubulin is not affected by the absence of TUBE1.**

Isolated elongating spermatids from *Tube1^Flox/Flox^* and *Tube1^GCKO/GCKO^* mice, immunolabeled for acetylated tubulin (red) and counterstained with DAPI (blue) (*n* = 3 animals/genotype). Scale bars: 2 μm. Results are from one experiment. Source data are available online for this figure.

