## [Peer Review File · EMBO Reports]

Epsilon tubulin is an essential determinant of microtubule-based structures in male germ cells

G Stathatos, D. Merriner, Anne O'Connor, Jennifer Zenker, Jessica Dunleavy, and Moira O'Bryan

Corresponding author(s): Moira O'Bryan (moira.obryan@unimelb.edu.au), Jessica Dunleavy (jessica.dunleavy@unimelb.edu.au)

Review Timeline:

Transfer Date:	23rd Sep 23
Editorial Decision:	4th Oct 23
Revision Received:	18th Feb 24
Editorial Decision:	5th Apr 24
Revision Received:	8th Apr 24
Accepted:	30th Apr 24

Editor: Deniz Senyilmaz Tiebe

Transaction Report: This manuscript was transferred to EMBO reports following peer review at Review Commons.

Review
COMMONS

Review #1

1. Evidence, reproducibility and clarity:

Evidence, reproducibility and clarity (Required)

The ZED (zeta-, epsilon-, and delta-) tubulins are important, yet understudied, members of the tubulin superfamily. Here, Stathatos et al. build upon previously published work and leverage their expertise to uncover the roles of epsilon-tubulin in mouse male germ cells. The authors create a germ cell-specific *Tube1* knockout mouse, using *Stra8-Cre*, which is active in spermatogonia before the meiotic divisions. The authors report that knockout of *Tube1* results in a range of defects during spermatogenesis, including: 1) a loss of male germ cells 2) sperm motility defects 3) abnormally shaped sperm heads 4) abnormal meiotic spindle morphology and abnormal centrosome numbers 5) some defects in sperm axoneme ultrastructure, 6) disrupted manchette migration 7) increased levels of katanin subunits at the manchette. Most of the experiments are convincing and well done, and based on this work, the authors propose a novel model for regulation of the manchette. I believe this work is of interest and should be published with revisions addressing the following major and minor comments.

****Major comment:****

1. A major claim of the paper is that epsilon-tubulin plays a different role within mammalian germ cells (abstract, line 22; p9, lines 167-168; p15 lines 315-316), because the *Tube1*GCKO/GCKO mice can form some sperm with relatively normal ciliary ultrastructure, whereas ciliates lacking epsilon-tubulin fail to form cilia. However, it's unclear whether the centrioles that templated these normal cilia were formed before or after epsilon-tubulin loss. Given that centrioles are inherited from one generation to the next, it's possible that the few normal cilia may be templated by relatively normal parental centrioles. These parental centrioles would have been present in spermatogonia prior to *Cre* expression/epsilon-tubulin deletion, and inherited by a fraction of sperm after the mitotic and meiotic divisions, resulting in sperm with normal ciliary ultrastructure. Other spermatocytes may have inherited centrioles formed in the absence of epsilon-tubulin, resulting in aberrant centrioles similar to those reported in human somatic cells, but these would not form any sperm flagella due to a loss of cell viability, as has been reported for acentriolar cells in a *p53*⁺ background. Underscoring this point, *Chlamydomonas* and human somatic mutant cells constitutively lack epsilon-tubulin. In these systems, the parental centrioles were diluted from the population over many cell divisions, and phenotypic

analysis would only include the centrioles that formed in the absence of epsilon-tubulin. To make their major claim, the authors need to demonstrate that the basal bodies of sperm flagella with normal ultrastructure were formed in the absence of epsilon-tubulin, and were not normal parental centrioles. Given the difficulty of this experiment, the authors may instead choose to remove their claim that epsilon-tubulin plays a different role within mammalian germ cells.

****Minor comments:****

1. The authors claim that because the TUBE1 knockout mouse have abnormal centrosome numbers during meiosis, there is a role for TUBE1 in suppressing supernumerary centriole formation. While this is one possibility, it's also possible that abnormal centrosome numbers arose as a result of cell division defects, especially because binucleate cells are present in mutants. The authors should edit the text to state that abnormal centrosome numbers may arise from either supernumerary centriole formation (by the templated or de novo pathways) or from failure to complete cell division.

OPTIONAL: to test these possibilities, the authors may choose to 1) count the number of centrioles in meiosis with two different centriole markers 2) stain for markers of mature centrioles, such as Cep164, to determine the number of parental centrioles.

2. In figure 5, based on quantification of fluorescence intensity, the authors conclude that loss of epsilon-tubulin results in an increase in the levels of KATNAL1, KATNAL2, and KATNB1. Given the inherent variability in immunofluorescence staining, the authors should at a minimum normalize their intensity measurements to those of an unrelated control protein stained in the same cell (ex: alpha-tubulin). It would be more convincing to quantify the levels of these proteins by Western blot (again, normalized to a control protein or to total cellular protein), which should be feasible given that the authors can isolate elongating spermatids.

2. Significance:

Significance (Required)

The strengths of this study lie in the careful phenotypic analysis of loss of epsilon-tubulin, which is well-done and very thorough. The limitations of the study are in interpretation of the results, specifically as relates to centriole formation, but can be addressed as indicated above. This work will be of interest to cell and developmental biologists, especially those interested in centrosomes, cilia, and spermatogenesis.

3. How much time do you estimate the authors will need to complete the suggested revisions:

Estimated time to Complete Revisions (Required)

(Decision Recommendation)

Between 1 and 3 months

Yes

Review #2

1. Evidence, reproducibility and clarity:

Evidence, reproducibility and clarity (Required)

The paper "Epsilon tubulin is an essential determinant of microtubule-based structures in male germ cells" provides the first insight into the essential function of Epsilon tubulin.

TUBE1 (epsilon tubulin) is a non-canonical tubulin localized at the pericentriolar material of somatic and germ cell centrosome. TUBE1 has been primarily studied in unicellular organisms and cell lines, and multiple studies have shown its role in ciliogenesis and flagellum formation. However, its role in mammals, specifically in fertility, is unknown. Here, Stathatos et al address the critical question of whether TUBE1 plays a role in mammalian spermatogenesis and fertility. The authors show by germline inactivation of TUBE1 that the mice lacking TUBE1 are sterile, defective in

meiosis, form abnormal manchette, and sperms are nonmotile. The authors further correlate that the TUBE1 functions together with KATNAL-1, KATNAL-1, and KATNB1, the microtubule severing protein. As little is known about the role of non-canonical tubulin like TUBE1 in fertility, this manuscript addresses a significant knowledge gap and generates an exciting hypothesis that TUBE1 regulates the KATNAL1-KATNB1 and KATNAL2-KATNB1 dynamic at manchette microtubules and perinuclear ring to control the manchette microtubule severing and migration.

Overall, the paper suggests that Epsilon tubulin is essential for multiple complex microtubule arrays, including the meiotic spindle, axoneme, and manchette; however, in the absence of Epsilon tubulin localization data, it is unclear which microtubule array is affected directly and which indirectly (e.g., is the axoneme defect is due to Epsilon tubulin in the axoneme or centriole?). In particular, it is interesting that in mice sperm, Epsilon tubulin is dispensable for centriole-mediated axoneme formation, its primary function in single-cell organisms (can this be due to compensation by the other tubulin isoforms?). Once the concerns below are resolved, the paper will be significant for the cytoskeleton and reproductive research fields.

****Major comment****

- Considering the suggested non-canonical function of Epsilon tubulin outside the centriole in mice sperm, it is critical to know the localization of the protein in spermatocytes during meiosis and spermatids during differentiation.
- Localization of Epsilon tubulin is needed to distinguish between mutant sperm cells and those that are not Epsilon tubulin mutants in the Tube1GCKO/GCKO mice. E.g., are the 28.07% of Tube1GCKO/GCKO tubules that showed a Sertoli cell only (SCO) phenotype the one where all the cells are mutants?

****Minor comment****

- It will help if the introduction summarizes the knowledge on Epsilon tubulin in spermatogenesis with emphasis on its localization and the method used to find the localization.
- The generated conditional germ cell-specific mutants are demonstrated by mRNA expression spermatocytes. It would help if DNA sequencing, western, and Immunohistochemical staining were used to show the gene and protein are affected.
- How many independent mutant animals were studied, and what was the efficiency of generating mutants with a complete mutant testis? From Fig 1c, it appears all mutants generated were total mutations in almost all cells - is this correct?
- Add a definition to "ZED-tubulins."
- "Suggesting a core TUBE1 function that can be supplemented by either z-tubulin or

TUBD1." Can you test what happens to mice Z and D tubulin isoforms in the mutant? Did their level increase in the centrioles? This is informative since there is no clear centriolar phenotype (other than centriole number that may be due to cell division failure) in mice spermatogenesis and the paper's central hypothesis in the introduction.

- From the paper, it is unclear if Epsilon tubulin is dispensable for centriole function only in sperm cells or if the same is true in mice somatic cells in vivo.
- Fig. S1 and other figures: "n {greater than or equal to} 3 samples/genotype" - this is unclear - please indicate the number of independent animals tested.
- "suppressing supernumerary centriole formation" is this due to access centriole formation or failed mitosis?
- The KATNAL1, KATNAL2, and KATNB1 staining in Fig 5 show multiple foci in the nucleus. Are these foci-specific staining or nonspecific? It is surprising to see such a large complex.
- How the staging of spermatids was performed needs to be explained in the method.
- The authors looked at the Metaphase stage cells to assess meiosis. It would be more interesting to look at the meiosis prophase I. Since the Stra8 acts very early leptotene stage, it would be interesting to see if meiosis is defective from the very beginning. Also, some suggest that the manchette is nucleated at the pachytene stage. Is the manchette defective from the very early stage of nucleation?
- Is the acetylation of manchette microtubules affected in the absence of TUBE1?

2. Significance:

Significance (Required)

Overall, the paper suggests that Epsilon tubulin is essential for multiple complex microtubule arrays, including the meiotic spindle, axoneme, and manchette; however, in the absence of Epsilon tubulin localization data, it is unclear which microtubule array is affected directly and which indirectly (e.g., is the axoneme defect is due to Epsilon tubulin in the axoneme or centriole?). In particular, it is interesting that in mice sperm, Epsilon tubulin is dispensable for centriole-mediated axoneme formation, its primary function in single-cell organisms (can this be due to compensation by the other tubulin isoforms?). Once the concerns are resolved, the paper will be significant for the cytoskeleton and reproductive research fields.

3. How much time do you estimate the authors will need to complete the suggested revisions:

Estimated time to Complete Revisions (Required)

(Decision Recommendation)

Between 1 and 3 months

Yes

Review #3

1. Evidence, reproducibility and clarity:

Evidence, reproducibility and clarity (Required)

In this study Stathatos et al looked at the function of epsilon tubulin (tubel), specifically in male germ cells. Previous work showed that tubel is an important member of the tubulin family but its function is more enigmatic compared to alpha, beta and gamma tubulin. The authors produced a mouse KO line of tubel and the data presented in this manuscript concerns the effects on spermatogenesis. They found that tubel is essential for multiple microtubule dependent functions, including meiosis, nuclear shaping and sperm motility.

The experimental part is of the highest quality and the manuscript is very well written. My only reservation with the manuscript is concerning the model proposed for manchette migration in the Discussion section (Figure 6). I find the proposed model highly speculative and pre-mature, not supported enough by data, as even admitted by the authors (lines 415-427). Having it as a figure and concluding remark gives it too much weight, my suggestion would be to remove figure 6 and tone down the discussion.

Minor points, a substantial percentage of sperm produced had a normal head shape in the KO (Figure 1I), which undermine the function of tube1 in nuclear shaping, the author should address this point in their manuscript. It is also curious whether there are phenotype in other tissues, can the authors comment on that?

2. Significance:

Significance (Required)

The observations reported are novel and will be highly valuable specifically for the sperm biology field but also very interesting to the microtubule field in general.

3. How much time do you estimate the authors will need to complete the suggested revisions:

Estimated time to Complete Revisions (Required)

(Decision Recommendation)

Less than 1 month

Yes

Revision Plan

Manuscript number: RC-2023-02111

Corresponding author(s): Moira O'Bryan

1. General Statements

We thank the Review Commons editor and the three reviewers for their overall positive responses in assessing this manuscript. Further, we appreciate and would like to reiterate the similarities across our three reviewers' comments regarding the significance of this work, where our examination of epsilon tubulin (TUBE1) during mammalian spermatogenesis will be valuable for both microtubule/cytoskeletal and developmental/ reproductive fields. Below, we have made point-by-point responses to the reviewers' comments, and outlined by the revisions we plan to make, or have made. All line numbers refer to the transferred manuscript file with tracked changes.

2. Description of the planned revisions

Reviewer 1: The authors claim that because the TUBE1 knockout mouse have abnormal centrosome numbers during meiosis, there is a role for TUBE1 in suppressing supernumerary centriole formation. While this is one possibility, it's also possible that abnormal centrosome numbers arose as a result of cell division defects, especially because binucleate cells are present in mutants. The authors should edit the text to state that abnormal centrosome numbers may arise from either supernumerary centriole formation (by the templated or de novo pathways) or from failure to complete cell division.

OPTIONAL: to test these possibilities, the authors may choose to 1) count the number of centrioles in meiosis with two different centriole markers 2) stain for markers of mature centrioles, such as Cep164, to determine the number of parental centrioles.

Response: This is a good point. Published data indicates that the Stra8-cre is active within a subset of undifferentiated spermatogonia, and in differentiated spermatogonia through to pre-leptotene spermatocytes (Sadate-Ngatchou et al., 2008). This raises the possibility that the increase in centriole numbers could be due to a failure to complete cell division if cre is active in mitotically active spermatogonia populations. The text has been appropriately modified in lines 207-209 and 352 to reflect these insights. We appreciate the Reviewer's optional suggestion to perform additional immunolabeling experiments and intend to examine the number of parental centrioles in spermatocytes during meiotic division using a marker of the distal or subdistal appendages. This data will be included in the final revised document.

Reviewer 2: Considering the suggested non-canonical function of Epsilon tubulin outside the centriole in mice sperm, it is critical to know the localization of the protein in spermatocytes during meiosis and spermatids during differentiation.

Revision Plan

Response: We agree with Reviewer 2 that determining the localization of TUBE1 in spermatocytes and spermatids would be desirable. However, we are yet to find an appropriate available antibody for this. We have previously assessed the specificity of a TUBE1 antibody (PA5-56917, Invitrogen), however, this antibody was not suitable for use in our mouse model. This aside, we have recently acquired a new TUBE1 antibody which we plan to evaluate its specificity during this revision period. If it appears to bind specifically to TUBE1, we will perform the requested localization experiments.

For clarification we have previously defined the location of TUBE1 in spermatids to the manchette and basal body in elongating spermatids (lines 72-74) (Dunleavy et al., 2017). Unfortunately, the antibody used in this study is now discontinued. The phenotypes observed as a consequence of TUBE1 loss of function in this study are, however, consistent with these patterns of localization.

Reviewer 2: Localization of Epsilon tubulin is needed to distinguish between mutant sperm cells and those that are not Epsilon tubulin mutants in the Tube1GCKO/GCKO mice. E.g., are the 28.07% of Tube1GCKO/GCKO tubules that showed a Sertoli cell only (SCO) phenotype the one where all the cells are mutants?

Response: As per our response to Reviewer 2's comment above, we plan to test a new TUBE1 antibody to determine TUBE1 localization in this model. Outlined in our response to Reviewer 2 below, we also plan to sequence DNA from mature epididymal sperm from our mutant mice to further confirm the deletion of *Tube1* exon 3.

Reviewer 2: The generated conditional germ cell-specific mutants are demonstrated by mRNA expression spermatocytes. It would help if DNA sequencing, western, and Immunohistochemical staining were used to show the gene and protein are affected.

Response: We thank Reviewer 2 for their suggestions. Should we successfully validate an appropriate TUBE1 antibody for use in our model, we will perform immunohistochemical staining during the revision process. Our qPCR results from purified spermatocytes however, strongly suggest that the *Tube1* gene is deleted in our model, noting that such purifications are on average 81% pure with the major contaminants being Sertoli cells and spermatids (Dunleavy et al., 2019). To further confirm the deletion of *Tube1* exon 3, we plan to sequence DNA from mature epididymal sperm from our mutant mice.

Reviewer 2: "Suggesting a core TUBE1 function that can be supplemented by either z-tubulin or TUBD1." Can you test what happens to mice Z and D tubulin isoforms in the mutant? Did their level increase in the centrioles? This is informative since there is no clear centriolar phenotype (other than centriole number that may be due to cell division failure) in mice spermatogenesis and the paper's central hypothesis in the introduction.

Revision Plan

Response: We appreciate this question by Reviewer 2. Zeta tubulin is not present in the mouse genome as outlined in our introduction (lines 38-39). We do acknowledge that exploring *Tubd1* will be informative in our mutant and thereby plan to examine its expression in round spermatids.

Reviewer 2: The authors looked at the Metaphase stage cells to assess meiosis. It would be more interesting to look at the meiosis prophase I. Since the Stra8 acts very early leptotene stage, it would be interesting to see if meiosis is defective from the very beginning. Also, some suggest that the manchette is nucleated at the pachytene stage. Is the manchette defective from the very early stage of nucleation?

Response: We thank Reviewer 2 for this suggestion. To this end, we plan to examine juvenile mouse testes at days 10 and 17 post-partum where leptotene and pachytene spermatocytes are the most mature germ cells respectively.

In regard to the Reviewer's comment of the manchette being nucleated in pachytene stage spermatocytes, we acknowledge that the precise mechanism of manchette nucleation has not been confirmed. We are aware of the alternative hypothesis introduced by Moreno and Schatten (2000), which postulates manchette microtubules may be nucleated prior to pachytene period, through their examination of bovine male germ cells. This hasn't, however, been supported by evidence and with more recent data, others have suggested that the manchette is nucleated at the centrosomal adjunct (Lehti and Sironen, 2016). Indeed, our unpublished data suggests this is the case (another study). Regardless, the origin of the microtubule seeds that ultimately extend to form the manchette is not relevant to the hypothesis we have proposed. As we note that in our manuscript and mouse model, manchettes appear to assemble normally in step 8 spermatids. Rather, their movement and disassembly is abnormal i.e. TUBE1 serves critical roles more manchette movement and disassembly rather than manchette formation.

Reviewer 2: Is the acetylation of manchette microtubules affected in the absence of TUBE1?

Response: Reviewer 2 raises an interesting question, which we plan to answer through immunolabeling of testis sections for acetylated tubulin in our control and mutant groups.

Reviewer 3: Minor points, a substantial percentage of sperm produced had a normal head shape in the KO (Figure 1I), which undermine the function of tube1 in nuclear shaping, the author should address this point in their manuscript. It is also curious whether there are phenotype in other tissues, can the authors comment on that?

Response: We thank Reviewer 3 for highlighting this point. As reported in Fig. 1I, 28.5% of sperm from *Tube1*^{GCKO/GCKO} epididymides have abnormal nuclear shape. This is a 4.4-fold increase over that seen in wild type sperm. These data clearly highlight the role of TUBE1 in defining nuclear morphology. Variations between cells does not undermine this conclusion. It

appears that prior to sperm release from the testis, the majority of TUBE1 null spermatids heads are abnormally shaped. However, in the epididymis there appears to be an increase in the proportion of normally shaped heads. We thus hypothesize that the high rates of spermiation failure in the TUBE1 null mice reflect the preferential removal of abnormally shaped sperm by Sertoli cells, thus enriching for normally shaped heads that are released. During the revision process, we will quantify the percentage of spermatids with normal versus abnormally shaped heads prior to spermiation in testis sections. All *Tube1* null mice were sterile.

To Reviewer 3's second point - we have not examined other tissues in this conditional male germ cell knockout mouse model, as the cre used in this manuscript is only expressed in the testis (Sadate-Ngatchou et al., 2008). Consistent with the specificity of the deletion, null male mice are overtly healthy, with the exception of male fertility, and exhibit normal body weight as detailed on line 123 and in Fig S1D.

3. Description of the revisions that have already been incorporated in the transferred manuscript

Reviewer 1: In figure 5, based on quantification of fluorescence intensity, the authors conclude that loss of epsilon-tubulin results in an increase in the levels of KATNAL1, KATNAL2, and KATNB1. Given the inherent variability in immunofluorescence staining, the authors should at a minimum normalize their intensity measurements to those of an unrelated control protein stained in the same cell (ex: alpha-tubulin). It would be more convincing to quantify the levels of these proteins by Western blot (again, normalized to a control protein or to total cellular protein), which should be feasible given that the authors can isolate elongating spermatids.

Response: We thank Reviewer 1 for this suggestion to better account for any potential variability between immunofluorescence staining in cells. In this instance, alpha-tubulin would be a related protein in our model, making it unsuitable for normalization - the longer manchette phenotypes in our mutant spermatids indicate more tubulin present in mutant cells. We have therefore normalized the fluorescence intensity in our cells to DNA content (DAPI staining). This has provided comparable results to our initial analysis, and we have edited our text accordingly at lines 303, 307-310, 563-564, 845, 850 and Fig. 5. We respectfully disagree that western blotting would be informative, as the point is that katanin proteins are accumulating abnormally on the elongating sperm manchette. This does not necessarily mean that overall katanin levels will be increased. This aside, given the low numbers of elongating spermatids in the *Tube1*^{GCKO/GCKO} mice, obtaining sufficient materials of western blotting is prohibitive. With the severity of germ cell loss indicated by our daily sperm production calculations, we predict the isolated spermatids of up to 5 *Tube1*^{GCKO/GCKO} animals would be required to make up one biological replicate. It would not be feasible to collect the large number of animals required for at least three biological replicates in the revision timeframe.

Reviewer 1: *A major claim of the paper is that epsilon-tubulin plays a different role within mammalian germ cells (abstract, line 22; p9, lines 167-168; p15 lines 315-316), because the Tube1GCKO/GCKO mice can form some sperm with relatively normal ciliary ultrastructure, whereas ciliates lacking epsilon-tubulin fail to form cilia. However, it's unclear whether the centrioles that templated these normal cilia were formed before or after epsilon-tubulin loss. Given that centrioles are inherited from one generation to the next, it's possible that the few normal cilia may be templated by relatively normal parental centrioles. These parental centrioles would have been present in spermatogonia prior to Cre expression/epsilon-tubulin deletion, and inherited by a fraction of sperm after the mitotic and meiotic divisions, resulting in sperm with normal ciliary ultrastructure. Other spermatocytes may have inherited centrioles formed in the absence of epsilon-tubulin, resulting in aberrant centrioles similar to those reported in human somatic cells, but these would not form any sperm flagella due to a loss of cell viability, as has been reported for acentriolar cells in a p53+ background. Underscoring this point, Chlamydomonas and human somatic mutant cells constitutively lack epsilon-tubulin. In these systems, the parental centrioles were diluted from the population over many cell divisions, and phenotypic analysis would only include the centrioles that formed in the absence of epsilon-tubulin. To make their major claim, the authors need to demonstrate that the basal bodies of sperm flagella with normal ultrastructure were formed in the absence of epsilon-tubulin, and were not normal parental centrioles. Given the difficulty of this experiment, the authors may instead choose to remove their claim that epsilon-tubulin plays a different role within mammalian germ cells.*

Response: The authors thank Reviewer 1 for their detailed input regarding TUBE1's centriolar importance across species. From their feedback, we recognize the need to modulate our interpretation of this result. We have also added a line to our manuscript highlighting that the normal axonemal structure observed may be due to the inheritance of normal centrioles (lines 328-329). We note however, that sperm produced within the null animals were immotile and that motility could not be recovered by the addition of exogenous ATP thus revealing that TUBE1 is required to form functional sperm tails.

Reviewer 2: *It will help if the introduction summarizes the knowledge on Epsilon tubulin in spermatogenesis with emphasis on its localization and the method used to find the localization.*

Response: We have modified the introduction accordingly in lines 72-73.

Reviewer 2: *How many independent mutant animals were studied, and what was the efficiency of generating mutants with a complete mutant testis? From Fig s1c, it appears all mutants generated were total mutations in almost all cells - is this correct?*

Response: We have updated the number of animals studied as per the comment below. Regarding the mutant status of our mouse model, we used *Stra8-Cre* which is active between early (postnatal day 3) spermatogonia to pre-leptotene spermatocytes (Sadate-Ngatchou et al.,

Revision Plan

2008) thus all spermatocytes, spermatids, and sperm will carry the deletion. As shown in Fig. S1C we measured a 90.1% reduction in *Tube1* mRNA expression from purified spermatocytes. As mentioned above, we note that the purified germ cells always contain a low percentage of contaminating cells. Using our optimized Staput method we obtain isolated germ cell populations of high purity, where in spermatocyte populations we calculate 19% contamination with other testicular cell types (e.g. somatic Sertoli/interstitial cells, spermatogonia, spermatids) (Dunleavy et al., 2019). We therefore believe the 9.9% *Tube1* mRNA expression detected in our *Tube1*^{GCKO/GCKO} group are the origin of that residual mRNA. We have included this information in the materials and methods section (lines 491-493).

Reviewer 2: Add a definition to "ZED-tubulins."

Response: A definition to the ZED-tubulins can be found on line 32.

Reviewer 2: From the paper, it is unclear if Epsilon tubulin is dispensable for centriole function only in sperm cells or if the same is true in mice somatic cells in vivo.

Response: In this study we have used a conditional male germ cell knockout mouse model to examine TUBE1's function specifically in male germ cells. As mentioned in our introduction, the function of TUBE1 has not been examined in murine somatic cells *in vivo* (lines 68-70). To avoid confusion, we have reiterated this point in lines 356-358 of our discussion.

Reviewer 2: Fig. S1 and other figures: "n {greater than or equal to} 3 samples/genotype" - this is unclear - please indicate the number of independent animals tested.

Response: We have modified the figure legends accordingly in lines 11-13 and 33-35 of the transferred supplementary information file and lines 787-788 and 810-811 of the transferred manuscript file.

Reviewer 2: "suppressing supernumerary centriole formation" is this due to excess centriole formation or failed mitosis?

Response: We acknowledge Reviewer 2's comment is similar to the comment made by Reviewer 1 above and note we have modified the associated text in lines 207-209 in response to the above comment.

Reviewer 2: The KATNAL1, KATNAL2, and KATNB1 staining in Fig 5 show multiple foci in the nucleus. Are these foci-specific staining or nonspecific? It is surprising to see such a large complex.

Response: As outlined in the materials and methods and the Fig. 5 legend, Fig. 5 displays three-dimensional (3D) z-stack images of whole elongating spermatids presented as 2D maximum intensity projections. The katanin subunit staining is around the nucleus rather than

inside of it, however the flattening of the image from 3D to 2D make the foci appear inside the nucleus. To clarify this, we have modified the Fig. 5 legend in lines 845 and 848.

Reviewer 2: How the staging of spermatids was performed needs to be explained in the method.

Response: We have included additional explanation the materials and methods section (lines 513-514).

Reviewer 3: The experimental part is of the highest quality and the manuscript is very well written. My only reservation with the manuscript is concerning the model proposed for manchette migration in the Discussion section (Figure 6). I find the proposed model highly speculative and pre-mature, not supported enough by data, as even admitted by the authors (lines 415-427). Having it as a figure and concluding remark gives it too much weight, my suggestion would be to remove figure 6 and tone down the discussion.

Response: The authors thank Reviewer 3 for their complimentary overview of our manuscript. We agree that some unanswered questions remain in our proposed model of manchette migration. This study has however, added several critical missing pieces. With respect, we prefer to keep Figure 6 in the manuscript as explaining manchette function to non-experts is very difficult without a visual aide. To ensure transparency with the audience that our model is indeed hypothetical, we have edited our discussion and Figure 6 legend to reflect this (lines 406, 417, 428, 435, 463, 860, 863, 869).

4. Description of analyses that authors prefer not to carry out

None

References

- DUNLEAVY, J. E., GRAFFEO, M., WOZNIAK, K., O'CONNOR, A. E., MERRINER, D. J., NGUYEN, J., SCHITTENHELM, R. B., HOUSTON, B. J. & O'BRYAN, M. K. 2022. Male mammalian meiosis and spermiogenesis is critically dependent on the shared functions of the katanins KATNA1 and KATNAL1. *bioRxiv*, 2022.11.11.516072.
- DUNLEAVY, J. E. M., O'CONNOR, A. E. & O'BRYAN, M. K. 2019. An optimised STAPUT method for the purification of mouse spermatocyte and spermatid populations. *Molecular Human Reproduction*.
- DUNLEAVY, J. E. M., OKUDA, H., O'CONNOR, A. E., MERRINER, D. J., O'DONNELL, L., JAMSAI, D., BERGMANN, M. & O'BRYAN, M. K. 2017. Katanin-like 2 (KATNAL2) functions in multiple aspects of haploid male germ cell development in the mouse. *PLOS Genetics*, 13.
- LEHTI, M. S. & SIRONEN, A. 2016. Formation and function of the manchette and flagellum during spermatogenesis. *Reproduction*, 151, R43-54.

Revision Plan

- MORENO, R. D. & SCHATTEN, G. 2000. Microtubule configurations and post-translational alpha-tubulin modifications during mammalian spermatogenesis. *Cell Motil Cytoskeleton*, 46, 235-46.
- SADATE-NGATCHOU, P. I., PAYNE, C. J., DEARTH, A. T. & BRAUN, R. E. 2008. Cre recombinase activity specific to postnatal, premeiotic male germ cells in transgenic mice. *Genesis*, 46, 738-42.

Dear Prof. O'Bryan,

Thank you for transferring your manuscript to EMBO Reports, which was previously reviewed at Review Commons.

Referees express interest in the proposed role of epsilon tubulin in regulation of complex microtubule arrays of mouse germ cells during spermatogenesis. However, they also raise concerns that need to be addressed to consider publication in EMBO Reports.

Having looked at all documents, we would like to invite you to submit a revised manuscript as in your revision plan. Please revise your manuscript with the understanding that the referee concerns (as in their reports) must be fully addressed and their suggestions taken on board. Please address all referee concerns in a complete point-by-point response. Acceptance of the manuscript will depend on a positive outcome of a second round of review. It is EMBO reports policy to allow a single round of major experimental revision only and acceptance or rejection of the manuscript will therefore depend on the completeness of your responses included in the next, final version of the manuscript.

We realize that it is difficult to revise to a specific deadline. In the interest of protecting the conceptual advance provided by the work, we recommend a revision within 3 months. Please discuss the revision progress ahead of this time with me if you require more time to complete the revisions, or if you have questions or comments regarding the revision (also by video chat).

1. A data availability section providing access to data deposited in public databases is missing (where applicable).
2. Your manuscript contains statistics and error bars based on $n=2$. Please use scatter plots in these cases.

You can submit the revision either as a Scientific Report or as a Research Article. For Scientific Reports, the revised manuscript can contain up to 5 main figures and 5 Expanded View figures, and it should not exceed 27000 characters. If the revision leads to a manuscript with more than 5 main figures it will be published as a Research Article. In this case the Results and Discussion section should be separate. If a Scientific Report is submitted, these sections have to be combined. This will help to shorten the manuscript text by eliminating some redundancy that is inevitable when discussing the same experiments twice. In either case, all materials and methods should be included in the main manuscript file.

4) a .docx formatted letter INCLUDING the reviewers' reports and your detailed point-by-point responses to their comments. As part of the EMBO publication's Transparent Editorial Process, EMBO reports publishes online a Review Process File (RPF) to accompany accepted manuscripts. This File will be published in conjunction with your paper and will include the referee reports, your point-by-point response and all pertinent correspondence relating to the manuscript.

<https://www.embopress.org/page/journal/14693178/authorguide#transparentprocess>

5) a complete author checklist, which you can download from our author guidelines <https://www.embopress.org/page/journal/14693178/authorguide>. Please insert information in the checklist that is also reflected in the manuscript. The completed author checklist will also be part of the RPF.

6) Please note that all corresponding authors are required to supply an ORCID ID for their name upon submission of a revised manuscript (). Please find instructions on how to link your ORCID ID to your account in our manuscript tracking system in our Author guidelines

Additional information on source data and instruction on how to label the files are available:
<https://www.embopress.org/page/journal/14693178/authorguide#sourcedata>

9) Our journal encourages inclusion of *data citations in the reference list* to directly cite datasets that were re-used and obtained from public databases. Data citations in the article text are distinct from normal bibliographical citations and should directly link to the database records from which the data can be accessed. In the main text, data citations are formatted as follows: "Data ref: Smith et al, 2001" or "Data ref: NCBI Sequence Read Archive PRJNA342805, 2017". In the Reference list, data citations must be labeled with "[DATASET]". A data reference must provide the database name, accession number/identifiers and a resolvable link to the landing page from which the data can be accessed at the end of the reference. Further instructions are available at <http://www.embopress.org/page/journal/14693178/authorguide#referencesformat>

12) Please also note our reference format:
<http://www.embopress.org/page/journal/14693178/authorguide#referencesformat>

I look forward to seeing a revised version of your manuscript when it is ready. Please let me know if you have questions or comments regarding the revision.

Kind regards,

Deniz Senyilmaz Tiebe

Deniz Senyilmaz Tiebe, PhD
Editor
EMBO Reports

Full Revision

Manuscript number: EMBOR-2023-58207V1 [RC-2023-02111]

Corresponding author(s): Moira O'Bryan

1. General Statements

We thank the Review Commons editor and the three reviewers for their overall positive responses in assessing this manuscript. Further, we appreciate and would like to reiterate the similarities across our three reviewers' comments regarding the significance of this work, where our examination of epsilon tubulin (TUBE1) during mammalian spermatogenesis will be valuable for both microtubule/cytoskeletal and developmental/ reproductive fields. Below, we have made point-by-point responses to the reviewers' comments and have highlighted changes in our manuscript in yellow in accordance with these comments and in line with the EMBO Reports format. Included in this, we have reformatted figure legends according to EMBO Reports style, specifically including the statistical test and replicate numbers for each figure panel. We have also adopted the EMBO Reports citation style and the expanded view (EV) figure format in place of supplementary figures. Line numbers refer to the revised manuscript file.

Reviewer #1 (*Evidence, reproducibility and clarity (Required)*):

The ZED (zeta-, epsilon-, and delta-) tubulins are important, yet understudied, members of the tubulin superfamily. Here, Stathatos et al. build upon previously published work and leverage their expertise to uncover the roles of epsilon-tubulin in mouse male germ cells. The authors create a germ cell-specific Tube1 knockout mouse, using Stra8-Cre, which is active in spermatogonia before the meiotic divisions. The authors report that knockout of Tube1 results in a range of defects during spermatogenesis, including: 1) a loss of male germ cells 2) sperm motility defects 3) abnormally shaped sperm heads 4) abnormal meiotic spindle morphology and abnormal centrosome numbers 5) some defects in sperm axoneme ultrastructure, 6) disrupted manchette migration 7) increased levels of katanin subunits at the manchette. Most of the experiments are convincing and well done, and based on this work, the authors propose a novel model for regulation of the manchette. I believe this work is of interest and should be published with revisions addressing the following major and minor comments.

Major comment:

1. A major claim of the paper is that epsilon-tubulin plays a different role within mammalian germ cells (abstract, line 22; p9, lines 167-168; p15 lines 315-316), because the Tube1GCKO/GCKO mice can form some sperm with relatively normal ciliary ultrastructure, whereas ciliates lacking epsilon-tubulin fail to form cilia. However, it's unclear whether the centrioles that templated these normal cilia were formed before or after epsilon-tubulin loss. Given that centrioles are inherited from one generation to the next, it's possible that the few normal cilia may be templated by relatively normal parental centrioles. These parental

centrioles would have been present in spermatogonia prior to Cre expression/epsilon-tubulin deletion, and inherited by a fraction of sperm after the mitotic and meiotic divisions, resulting in sperm with normal ciliary ultrastructure. Other spermatocytes may have inherited centrioles formed in the absence of epsilon-tubulin, resulting in aberrant centrioles similar to those reported in human somatic cells, but these would not form any sperm flagella due to a loss of cell viability, as has been reported for acentriolar cells in a p53+ background. Underscoring this point, Chlamydomonas and human somatic mutant cells constitutively lack epsilon-tubulin. In these systems, the parental centrioles were diluted from the population over many cell divisions, and phenotypic analysis would only include the centrioles that formed in the absence of epsilon-tubulin. To make their major claim, the authors need to demonstrate that the basal bodies of sperm flagella with normal ultrastructure were formed in the absence of epsilon-tubulin, and were not normal parental centrioles. Given the difficulty of this experiment, the authors may instead choose to remove their claim that epsilon-tubulin plays a different role within mammalian germ cells.

The authors thank Reviewer 1 for their detailed input regarding TUBE1's centriolar importance across species. From their feedback, we recognize the need to modulate our interpretation of this result and have removed statements on lines 23-24 and 197-198 regarding a potential difference in TUBE1 function between mammalian germ cells compared to unicellular flagellates/ciliates. We have also added a line to our manuscript highlighting that the normal axonemal structure observed may be due to the inheritance of normal centrioles (lines 365-370). We note however, that sperm produced within the null animals were immotile and that motility could not be recovered by the addition of exogenous ATP thus revealing that TUBE1 is required to form functional sperm tails (line 370-372).

Minor comments:

1. The authors claim that because the TUBE1 knockout mouse have abnormal centrosome numbers during meiosis, there is a role for TUBE1 in suppressing supernumerary centriole formation. While this is one possibility, it's also possible that abnormal centrosome numbers arose as a result of cell division defects, especially because binucleate cells are present in mutants. The authors should edit the text to state that abnormal centrosome numbers may arise from either supernumerary centriole formation (by the templated or de novo pathways) or from failure to complete cell division.

OPTIONAL: to test these possibilities, the authors may choose to 1) count the number of centrioles in meiosis with two different centriole markers 2) stain for markers of mature centrioles, such as Cep164, to determine the number of parental centrioles.

We thank Reviewer 1 for their suggestion and have modified our statement (lines 239-243 and 391-392) to reflect that the increase in centriole numbers could be due to a combination of centriole and division defects. We appreciate the Reviewer's optional suggestion to perform additional immunolabeling experiments. During this revision process we revisited data from Wellard et al. (2021) (EMBO Reports) that showed that in addition to parental centrioles,

spermatocyte daughter centrioles also have distal appendages. Regardless, we did test several centriole-associated antibodies alongside our centrin antibody for the dual labelling of centrioles (ninein, Cep164, Cep170) however, they did not consistently show centriole labelling in wild-type metaphase isolated spermatocytes. We instead examined the number of centrosomes present in metaphase spermatocytes by dual labeling of centrin and gamma tubulin. We found a higher proportion of metaphase spermatocytes with more or less than 2 centrosomes from *Tube1*^{GCKO/GCKO} mice (46.7%) compared to that in *Tube1*^{Flox/Flox} populations (5.6%). This work is included in Fig EV3D and lines 235-238, methods section lines 599-600, and 1063-1065.

2. In figure 5, based on quantification of fluorescence intensity, the authors conclude that loss of epsilon-tubulin results in an increase in the levels of KATNAL1, KATNAL2, and KATNB1. Given the inherent variability in immunofluorescence staining, the authors should at a minimum normalize their intensity measurements to those of an unrelated control protein stained in the same cell (ex: alpha-tubulin). It would be more convincing to quantify the levels of these proteins by Western blot (again, normalized to a control protein or to total cellular protein), which should be feasible given that the authors can isolate elongating spermatids.

We thank Reviewer 1 for this suggestion to better account for any potential variability between immunofluorescence staining in cells. We agree, alpha-tubulin is not an appropriate 'house-keeping' protein given its potential link to TUBE1 biology and the longer manchette phenotypes in our mutant spermatids. We have therefore normalized the fluorescence intensity in our cells to DNA content (DAPI staining). This has provided comparable results to our initial analysis, and we have edited our text accordingly at lines 342-343, 347, 348-350, 621-627 and Fig 5. We have also edited the Fig 5 legend (lines 951-981) to include this information, which is now formatted in the EMBO Reports style. We respectfully disagree that western blotting would be informative, as the point is that katanin proteins are accumulating specifically on the elongating sperm manchette. This does not necessarily mean that overall katanin levels will be increased i.e. both quantity and context are important. This aside, given the low numbers of elongating spermatids in the *Tube1*^{GCKO/GCKO} mice, obtaining sufficient materials of western blotting is prohibitive. With the severity of germ cell loss indicated by our daily sperm production calculations, we predict the isolated spermatids of up to 5 *Tube1*^{GCKO/GCKO} animals would be required to make up one biological replicate – noting that null males are sterile, heterozygote crosses are required, and assuming 1 male pup of the correct genotype is produced per litter (assuming a litter of 8 pups on average), this equates to 5 litters per biological replicate. As animals need to be aged to 10 weeks of age prior to analysis (3 weeks gestation + 10 weeks aging), it was not feasible to collect the large number of animals required for at least three biological replicates in the revision timeframe.

Reviewer #1 (Significance (Required)):

The strengths of this study lie in the careful phenotypic analysis of loss of epsilon-tubulin, which is well-done and very thorough. The limitations of the study are in interpretation of the results, specifically as relates to centriole formation, but can be addressed as indicated

above. This work will be of interest to cell and developmental biologists, especially those interested in centrosomes, cilia, and spermatogenesis.

Once again, we thank Reviewer 1 for the time taken to review our manuscript.

Reviewer #2 (*Evidence, reproducibility and clarity (Required)*):

The paper "Epsilon tubulin is an essential determinant of microtubule-based structures in male germ cells" provides the first insight into the essential function of Epsilon tubulin.

TUBE1 (epsilon tubulin) is a non-canonical tubulin localized at the pericentriolar material of somatic and germ cell centrosome. TUBE1 has been primarily studied in unicellular organisms and cell lines, and multiple studies have shown its role in ciliogenesis and flagellum formation. However, its role in mammals, specifically in fertility, is unknown. Here, Stathatos et al address the critical question of whether TUBE1 plays a role in mammalian spermatogenesis and fertility. The authors show by germline inactivation of TUBE1 that the mice lacking TUBE1 are sterile, defective in meiosis, form abnormal manchette, and sperms are nonmotile. The authors further correlate that the TUBE1 functions together with KATNAL-1, KATNAL-1, and KATNB1, the microtubule severing protein. As little is known about the role of non-canonical tubulin like TUBE1 in fertility, this manuscript addresses a significant knowledge gap and generates an exciting hypothesis that TUBE1 regulates the KATNAL1-KATNB1 and KATNAL2-KATNB1 dynamic at manchette microtubules and perinuclear ring to control the manchette microtubule severing and migration.

Overall, the paper suggests that Epsilon tubulin is essential for multiple complex microtubule arrays, including the meiotic spindle, axoneme, and manchette; however, in the absence of Epsilon tubulin localization data, it is unclear which microtubule array is affected directly and which indirectly (e.g., is the axoneme defect is due to Epsilon tubulin in the axoneme or centriole?). In particular, it is interesting that in mice sperm, Epsilon tubulin is dispensable for centriole-mediated axoneme formation, its primary function in single-cell organisms (can this be due to compensation by the other tubulin isoforms?). Once the concerns below are resolved, the paper will be significant for the cytoskeleton and reproductive research fields.

Major comment

- Considering the suggested non-canonical function of Epsilon tubulin outside the centriole in mice sperm, it is critical to know the localization of the protein in spermatocytes during meiosis and spermatids during differentiation.

We agree with Reviewer 2 that determining the localization of TUBE1 in spermatocytes and spermatids would be desirable, however, we have not found an appropriate antibody for this. We have assessed the specificity of a TUBE1 antibody validated in a human cell line (PA5-56917, Invitrogen) and a monoclonal TUBE1 antibody (BSM-33314M, Bioss), however, both bound non-specifically to other proteins.

For clarification we have previously defined the location of TUBE1 in spermatids to the manchette and basal body in elongating spermatids (lines 64-66 and 83-85) (Dunleavy et al., 2017). Unfortunately, the antibody used in this study is now discontinued. The phenotypes observed as a consequence of TUBE1 loss of function in this study are, however, consistent with these patterns of localization.

We also respectfully disagree that the potential for compensation needs to be clarified to reach significance. The manuscript already contains a huge amount of new information including mechanistic data. To produce e.g. double knockout mice, which is perhaps what is being suggested, and characterize them to the level outlined here, would constitute a tripling of the content and work included here already. Mouse models of infertility are exceptionally good for defining biology in context. Unfortunately, however, they are extremely laborious and not amendable to technologies often used in cell lines such as transfection or rapid CRISPR deletion.

- Localization of Epsilon tubulin is needed to distinguish between mutant sperm cells and those that are not Epsilon tubulin mutants in the Tube1GCKO/GCKO mice. E.g., are the 28.07% of Tube1GCKO/GCKO tubules that showed a Sertoli cell only (SCO) phenotype the one where all the cells are mutants?

As per our response to Reviewer 2's comment above, we were unable to source an appropriate TUBE1 antibody in this manuscript. We note, however, that immunolocalization of TUBE1 in sperm would not be informative, because like many other proteins that are shed prior to spermiation, TUBE1 is not present in wild-type mature epididymal sperm as assessed by mass spectrometry (Skerrett-Byrne et al., 2022). Equally, protein and mRNA products are shared between sister germ cells via intracellular bridges meaning that individual sperm can be genetically null but heterozygous at a protein level.

Moreover, as detailed in our response below, our data strongly supports *Tube1* being deleted from all spermatocytes, spermatids, and sperm. Given that in our model TUBE1 is still present in the somatic cells of the testis and should be present in almost all spermatogonia cells (noting that the *Stra8-Cre* has some activity in a subset of undifferentiated spermatogonia (Sadate-Ngatchou et al., 2008)), the SCO seminiferous tubule presentation is likely a secondary phenotype to germ cell abnormalities and a result of premature germ cell sloughing. Regardless, we accept the possibility that the SCO seminiferous tubule phenotypes may be a result of the premature deletion of *Tube1* from spermatogonial stem cells, resulting in a more severe disruption to seminiferous epithelium integrity. On a technical note, SCO tubules do not have germ cells in them, so it is not possible to test for mutant cells.

Minor comment

- It will help if the introduction summarizes the knowledge on Epsilon tubulin in spermatogenesis with emphasis on its localization and the method used to find the localization.

Full Revision

We have modified the introduction accordingly in lines 83-85.

- The generated conditional germ cell-specific mutants are demonstrated by mRNA expression spermatocytes. It would help if DNA sequencing, western, and immunohistochemical staining were used to show the gene and protein are affected.

We thank Reviewer 2 for their suggestions. As indicated above, the lack of appropriate antibody against TUBE1 means we cannot examine its protein level/expression via western blot or immunohistochemical staining. As indicated by the reviewer however, our qPCR results from purified spermatocytes strongly suggest that the *Tube1* gene is deleted in our model (90.1% mRNA reduction), noting that such preparations are on average 81% pure with the major contaminants being Sertoli cells and spermatids (Dunleavy et al., 2019). This is also consistent with our data showing the severe abnormalities observed in the sperm from *Tube1*^{GCKO/GCKO} mice, compared to controls, for example the 91.6% reduction in epididymal sperm number, and of these sperm that are present in the epididymis, the 97.7% reduction in their functional progressive motility.

Given the strength of this data, germ cell populations from mice available in the review timeframe were prioritized for other experiments as opposed to sequencing. Due to the presence of sloughed somatic cells and germ cells (that would include types prior to deletion) in the epididymis, sequencing of backflushed epididymal samples would not be as accurate.

- How many independent mutant animals were studied, and what was the selfishness of generating mutants with a complete mutant testis? From Fig s1c, it appears all mutants generated were total mutations in almost all cells - is this correct?

We have updated the number of animals studied in our figure legends as per the comment below. Regarding the mutant status of our mouse model, we used *Stra8-Cre* which is active between early (postnatal day 3) spermatogonia to pre-leptotene spermatocytes (Sadate-Ngatchou et al., 2008) thus all spermatocytes, spermatids, and sperm should carry the deletion. Somatic cells remain wild type. As shown in Fig. S1C we measured a 90.1% reduction in *Tube1* mRNA expression from purified spermatocytes. As mentioned above, we note that the purified germ cells always contain a low percentage of contaminating cells, where in spermatocyte populations we calculate 19% contamination with other testicular cell types (e.g. somatic Sertoli/interstitial cells, spermatogonia, spermatids) (Dunleavy et al., 2019). We therefore believe the 9.9% *Tube1* mRNA expression detected in our *Tube1*^{GCKO/GCKO} group originate from that residual spermatogonia and somatic cell mRNA. We have included this information in lines 135-137 and the materials and methods section (lines 535-537).

- Add a definition to "ZED-tubulins."

A definition to the ZED-tubulins can be found on lines 35-36.

- "Suggesting a core TUBE1 function that can be supplemented by either z-tubulin or TUBD1." Can you test what happens to mice Z and D tubulin isoforms in the mutant? Did their level increase in the centrioles? This is informative since there is no clear centriolar phenotype (other than centriole number that may be due to cell division failure) in mice spermatogenesis and the paper's central hypothesis in the introduction.

We appreciate this question by Reviewer 2. Zeta tubulin is not present in the mouse genome as outlined in our introduction (lines 43-44). In terms of testing what happens to TUBD1 in our mutant model, we examined *Tubd1* mRNA expression in round spermatids and found at 56% reduction of *Tubd1* mRNA in *Tube1* null round spermatids when compared to control round spermatids. This data will be important for future work and is included in Figure EV2G, with accompanying information in lines 187-190, 433-434, 554 and 557-563.

- From the paper, it is unclear if Epsilon tubulin is dispensable for centriole function only in sperm cells or if the same is true in mice somatic cells *in vivo*.

In this study we have used a conditional male germ cell knockout mouse model to examine TUBE1's function specifically in male germ cells. As mentioned in our introduction, the function of TUBE1 has not been examined in murine somatic cells *in vivo* (lines 78-80). To avoid confusion, we have reiterated this point in lines 395-397 of our discussion.

- Fig. S1 and other figures: "*n* {greater than or equal to} 3 samples/genotype" - this is unclear - please indicate the number of independent animals tested.

We have modified the figure legends accordingly in each figure of the revised manuscript, including our expanded view figures (previously supplementary figures) lines 813-981, 1009-1076. We note that each figure legend has also changed its formatting style according to EMBO Reports guidelines.

- "suppressing supernumerary centriole formation" is this due to access centriole formation or failed mitosis?

We acknowledge Reviewer 2's comment is similar to the comment made by Reviewer 1 above and note we have modified the associated text in lines 239-243 in response to the above comment.

- The *KATNAL1*, *KATNAL2*, and *KATNB1* staining in Fig 5 show multiple foci in the nucleus. Are these foci-specific staining or nonspecific? It is surprising to see such a large complex.

As outlined in the materials and methods and the Fig. 5 legend, Fig. 5 displays three-dimensional (3D) z-stack images of whole elongating spermatids presented as 2D maximum intensity projections. The katanin subunit staining is around the nucleus rather than inside of it, however the flattening of the image from 3D to 2D make the foci appear inside the nucleus. To

clarify this, we have modified the Fig 5 legend as mentioned above, with this information now included in lines 953-954.

- How the staging of spermatids was performed needs to be explained in the method.

We have included additional explanation the materials and methods section (lines 567-568).

- The authors looked at the Metaphase stage cells to assess meiosis. It would be more interesting to look at the meiosis prophase I. Since the Stra8 acts very early leptotene stage, it would be interesting to see if meiosis is defective from the very beginning. Also, some suggest that the manchette is nucleated at the pachytene stage. Is the manchette defective from the very early stage of nucleation?

We thank Reviewer 2 for this suggestion. We have examined the beginning of meiosis prophase I in two measures, which are presented in Figure EV3A and C. In the first measure, we counted the number of leptotene and pachytene cells present in Stage IX seminiferous tubule sections in both *Tube1^{Flox/Flox}* and *Tube1^{GCKO/GCKO}* groups and found they were comparable between genotypes. In the second measure, we immunolabeled isolated leptotene spermatocytes with SYCP3 (to identify leptotene) and β -tubulin (to examine microtubule architecture) and found the loss of TUBE1 had no discernible effect on microtubule architecture at the beginning of meiosis I. Text has been added at lines 202, 204-206, 228-229, materials and methods lines 603-604, and EV figure legend lines 1052-1055 and 1059-1062 to reflect these results.

In regard to the Reviewer's comment of the manchette being nucleated in pachytene stage spermatocytes, we acknowledge that the precise mechanism of manchette nucleation has not been confirmed. We are aware of the alternative hypothesis introduced by Moreno and Schatten (2000), which postulates manchette microtubules may be nucleated prior to pachytene period, through their examination of bovine male germ cells. This hasn't, however, been supported by evidence and with more recent data, others have suggested that the manchette is nucleated at the centrosomal adjunct (Lehti and Sironen, 2016). Indeed, our unpublished data suggests this is the case (another study). Regardless, the origin of the microtubule seeds that ultimately extend to form the manchette is not relevant to the hypothesis we have proposed. As we note that in our manuscript and mouse model, manchettes appear to assemble normally in step 8 spermatids. Rather, their movement and disassembly is abnormal i.e. TUBE1 serves critical roles to allow manchette movement and disassembly rather than manchette formation.

- Is the acetylation of manchette microtubules affected in the absence of TUBE1?

We have included immunofluorescence data of isolated elongating spermatids from *Tube1^{Flox/Flox}* and *Tube1^{GCKO/GCKO}* testes with acetylated tubulin staining at their manchettes (Figure EV5), where acetylation doesn't appear to be affected in the absence of TUBE1, and have updated our text (lines 300-303, materials and methods lines 598-599 and EV figure legend lines 1073-1076) to reflect this.

Reviewer #2 (Significance (Required)):

Overall, the paper suggests that Epsilon tubulin is essential for multiple complex microtubule arrays, including the meiotic spindle, axoneme, and manchette; however, in the absence of Epsilon tubulin localization data, it is unclear which microtubule array is affected directly and which indirectly (e.g., is the axoneme defect is due to Epsilon tubulin in the axoneme or centriole?). In particular, it is interesting that in mice sperm, Epsilon tubulin is dispensable for centriole-mediated axoneme formation, its primary function in single-cell organisms (can this be due to compensation by the other tubulin isoforms?). Once the concerns are resolved, the paper will be significant for the cytoskeleton and reproductive research fields.

Once again, we thank Reviewer 2 for the time taken to review our manuscript.

Reviewer #3 (Evidence, reproducibility and clarity (Required)):

In this study Stathatos et al looked at the function of epsilon tubulin (*tube1*), specifically in male germ cells. Previous work showed that *tube1* is an important member of the tubulin family but its function is more enigmatic compared to alpha, beta and gamma tubulin. The authors produced a mouse KO line of *tube1* and the data presented in this manuscript concerns the effects on spermatogenesis. They found that *tube1* is essential for multiple microtubule dependent functions, including meiosis, nuclear shaping and sperm motility.

The experimental part is of the highest quality and the manuscript is very well written.

My only reservation with the manuscript is concerning the model proposed for manchette migration in the Discussion section (Figure 6). I find the proposed model highly speculative and pre-mature, not supported enough by data, as even admitted by the authors (lines 415-427). Having it as a figure and concluding remark gives it too much weight, my suggestion would be to remove figure 6 and tone down the discussion.

Response: The authors thank Reviewer 3 for their complimentary overview of our manuscript. We agree that some unanswered questions remain in our proposed model of manchette migration. This study has however, added several critical missing pieces. With respect, we prefer to keep Figure 6 in the manuscript as explaining manchette function to non-experts is very difficult without a visual aide. To ensure transparency with the audience that our model is indeed hypothetical, we have edited our discussion and Figure 6 legend to reflect this (lines 449, 461, 470, 478, 983, 986, 992).

*Minor points, a substantial percentage of sperm produced had a normal head shape in the KO (Figure 11), which undermine the function of *tube1* in nuclear shaping, the author should address this point in their manuscript. It is also curious whether there are phenotype in other tissues, can the authors comment on that?*

We thank Reviewer 3 for highlighting this point. As reported in Fig. 11, 28.5% of sperm from *Tube1*^{GCKO/GCKO} epididymides have abnormal nuclear shape. This is a 4.4-fold increase over that

seen in wild type sperm. These data clearly highlight the role of TUBE1 in defining nuclear morphology. Variations between cells does not undermine this conclusion. It appears that prior to sperm release from the testis, the majority of TUBE1 null spermatids heads are abnormally shaped. However, in the epididymis there appears to be an increase in the proportion of normally shaped heads.

We therefore quantified the percentage of elongating spermatids within testis sections at a specific spermatogenic stage (Stage XI) that had normal versus abnormal (hyper-elongated) sperm heads prior to spermiation. Here, we calculated that 50.6% of spermatids were hyper-elongated in *Tube1*^{GCKO/GCKO} testes, which is an 8.9-fold increase to that observed in the controls. We note, that this analysis does not allow a full assessment of nuclear shape as it relies on a single slice/section of the head, as such, the numbers reported here are likely an under-estimate of the real situation. We have included this data in Figure EV4 and accompanying lines 182-187, 293-294, materials and methods lines 598-599, and EV figure lines 1067-1071.

We do, however, accept the reviewer's point and have modified the text in lines 312-313, to indicate that TUBE1 facilitates sperm head shaping, rather than it being a key determinant. Given the absolute requirement for TUBE1 to achieve functional sperm motility, we don't believe this undermines the conclusions of the paper.

To Reviewer 3's second point - we have not examined other tissues in this conditional male germ cell knockout mouse model, as the Cre used in this manuscript is only expressed in the testis (Sadate-Ngatchou et al., 2008). Consistent with the specificity of the deletion, null male mice are overtly healthy, with the exception of male fertility, and exhibit normal body weight as detailed on line 138-139 and in Fig EV1D.

Reviewer #3 (Significance (Required)):

The observations reported are novel and will be highly valuable specifically for the sperm biology field but also very interesting to the microtubule field in general.

Once again, we thank Reviewer 3 for the time taken to review our manuscript.

References

- DUNLEAVY, J. E. M., O'CONNOR, A. E. & O'BRYAN, M. K. 2019. An optimised STAPUT method for the purification of mouse spermatocyte and spermatid populations. *Molecular Human Reproduction*.
- DUNLEAVY, J. E. M., OKUDA, H., O'CONNOR, A. E., MERRINER, D. J., O'DONNELL, L., JAMSAI, D., BERGMANN, M. & O'BRYAN, M. K. 2017. Katanin-like 2 (KATNAL2) functions in multiple aspects of haploid male germ cell development in the mouse. *PLOS Genetics*, 13.
- LEHTI, M. S. & SIRONEN, A. 2016. Formation and function of the manchette and flagellum during spermatogenesis. *Reproduction*, 151, R43-54.
- MORENO, R. D. & SCHATTEN, G. 2000. Microtubule configurations and post-translational alpha-tubulin modifications during mammalian spermatogenesis. *Cell Motil Cytoskeleton*, 46, 235-46.
- SADATE-NGATCHOU, P. I., PAYNE, C. J., DEARTH, A. T. & BRAUN, R. E. 2008. Cre recombinase activity specific to postnatal, premeiotic male germ cells in transgenic mice. *Genesis*, 46, 738-42.
- SKERRETT-BYRNE, D. A., ANDERSON, A. L., BROMFIELD, E. G., BERNSTEIN, I. R., MULHALL, J. E., SCHJENKEN, J. E., DUN, M. D., HUMPHREY, S. J. & NIXON, B. 2022. Global profiling of the proteomic changes associated with the post-testicular maturation of mouse spermatozoa. *Cell Reports*, 41.
- WELLARD, S. R., ZHANG, Y., SHULTS, C., ZHAO, X., MCKAY, M., MURRAY, S. A. & JORDAN, P. W. 2021. Overlapping roles for PLK1 and Aurora A during meiotic centrosome biogenesis in mouse spermatocytes. *EMBO reports*, 22, e51023.

Dear Prof. O'Bryan,

Thank you for submitting your revised manuscript. It has now been seen by two of the original referees.

As you can see, the referees find that the study is significantly improved during revision and recommend publication. However, I need you to address the points below before I can accept the manuscript.

- Please address the remaining minor concerns of referee #1.
- Please remove the Author Contributions section from the manuscript.
- Please rename the Competing Interests section as "Disclosure Statement and Competing Interests".
- Funding information should be included in the Acknowledgements section.
- We note the following regarding figure callouts: every occurrence of the figure callout with prefix "S" need to be updated to the correct figure callout in the manuscript since there are only main and EV figures: S1A, S1B, S1C, S1D, S1G, S2A, S2B, S2 C and D, S2D, S2E, S2F.
- Source data of EV figures need to be grouped and uploaded as one zip folder.
- Our production/data editors have asked you to clarify several points in the figure legends:
 - o Please note that the legends for figures 5b-g are not provided in the sequential manner (legend for figure 5c, e, g are provided before legend of figure 5b, d, f). This needs to be rectified.

Thank you again for giving us to consider your manuscript for EMBO Reports, I look forward to your minor revision.

Kind regards,

Deniz Senyilmaz Tiebe

--

Deniz Senyilmaz Tiebe, PhD
Editor
EMBO Reports

Referee #1:

The authors have made many changes to the manuscript, including the addition of new figures and analyses. They have addressed most of my comments on the original manuscript. However, lines 164-166 still state that "Sperm present in the Tube1GCKO/GCKO epididymis possessed tails, indicating that in contrast to motile flagella in *C. reinhardtii* and *P. tetraurelia*, TUBE1 is not required to form mammalian sperm flagella per se." It is unclear whether the sperm tails formed in the Tube1GCKO/GCKO mice were templated by centrioles that were formed before or after epsilon-tubulin loss. The authors should revise lines 164-166 in the same way that they have for similar statements in the text (e.g. lines 198-199).

Overall, the manuscript demonstrates strong evidence for the conclusions that are drawn, is of appropriate length and format, demonstrates physiological and functional relevance, and the careful phenotypic analysis of loss of epsilon-tubulin is well-done and very thorough. With the additional single revision above, I believe this paper is suitable for publication in EMBO reports.

Referee #2:

The authors adequately addressed all my comments. I support the publication of this manuscript.

All editorial and formatting issues were resolved by the authors.

Dear Prof. O'Bryan,

Thank you for submitting your revised manuscript. I have now looked at everything and all is fine. Therefore, I am very pleased to accept your manuscript for publication in EMBO Reports.

Congratulations on a nice work!

Before we can transfer your manuscript to our production team, I need your input on the following. When your synopsis image is resized according our format requirements, the labels are too small to read (please see attached). Please provide a synopsis image with larger labels. Thank you.

Kind regards,

Deniz Senyilmaz Tiebe

--

Deniz Senyilmaz Tiebe, PhD

Editor

EMBO Reports

--
